# ULTRA-SPARSE MEMORY NETWORK

**Zihao Huang,*  Qiyang Min,*  Hongzhi Huang,*  Defa Zhu, Yutao Zeng, Ran Guo, Xun Zhou**
Seed-Foundation-Model Team, ByteDance
`{huangzihao.notabot,minqiyang,huanghongzhi.51,zhudefa,`
`yutao.zeng,guoran.94,zhouxun}@bytedance.com`

## ABSTRACT

It is widely acknowledged that the performance of Transformer models is logarithmically related to their number of parameters and computational complexity. While approaches like Mixture of Experts (MoE) decouple parameter count from computational complexity, they still face challenges in inference due to high memory access costs. This work introduces UltraMem, incorporating large-scale, ultra-sparse memory layer to address these limitations. Our approach significantly reduces inference latency while maintaining model performance. We also investigate the scaling laws of this new architecture, demonstrating that it not only exhibits favorable scaling properties but outperforms MoE. In experiments, the largest UltraMem we train has **20 million** memory slots. The results show that our method achieves state-of-the-art inference speed and model performance within a given computational budget, paving the way for billions of slots or experts.

## 1 INTRODUCTION

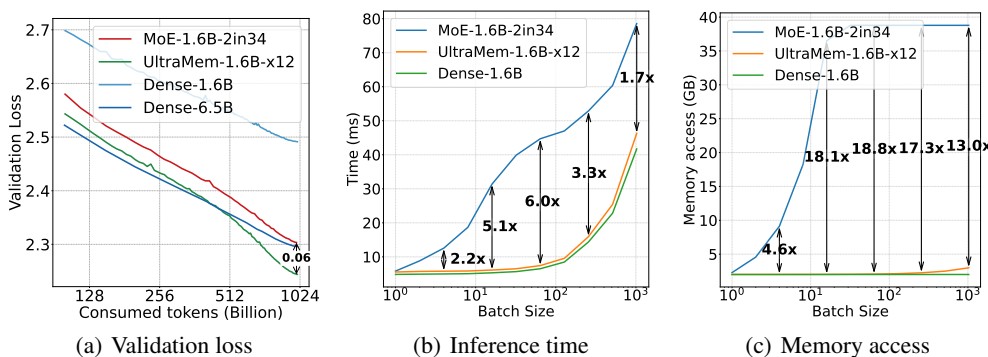

(a) Validation loss   (b) Inference time   (c) Memory access

Figure 1: We ensured that three models have the same computation, and MoE and UltraMem have the same parameters. The x-axis is plotted on a logarithmic scale. In (b) and (c), the sequence length is 1 because during decoding time, we can only predict one token at a time, and the key/value cache length is 2048. The experiments in (b) and (c) are conducted on the A100-SXM-80GB.

Recent advancements in natural language processing (NLP), driven by Large Language Models (LLMs) (Radford et al., 2019; Brown, 2020), require exponentially more computational resources as they scale, posing challenges in resource-limited environments like real-time applications. To address computational issues, the Mixture of Experts (MoE)(Fedus et al., 2022; Jiang et al., 2024) and Product Key Memory (PKM)(Lample et al., 2019) have been introduced. MoE selectively activates parameters, boosting training efficiency but impairing inference time due to increased memory access. PKM maintains consistent memory access with fewer value embeddings but its performance is significantly worse than MoE.

As shown in Figure 1(b), an MoE model, despite having the same computational cost and twelve times more parameters than a dense model, runs 2 to 6 times slower in inference, varying by batch size.

---

*Equal contribution.

This slowdown, as depicted in Figure 1(c), stems from high memory access demands, highlighting its inefficiency in inference scenarios. The primary challenge is how to match or even surpass the effectiveness of the MoE model while maintaining memory access levels comparable to those of dense models.

In this paper, we introduce UltraMem, an architecture that builds upon and extends the concepts from PKM. UltraMem incorporates large-scale, ultra-sparse memory layers that significantly enhance computational efficiency and reduce inference latency while maintaining or even improving model performance across various benchmarks. This architecture not only supports the deployment of highly effective language models in resource-constrained environments but also opens up new avenues for constructing even larger models without the previously associated prohibitive costs.

In summary, we make the following contributions:

1. UltraMem is greatly enhanced compared to PKM, and outperforms MoE at same scale. Compared to PKM, UltraMem truly possesses the prerequisites for training large-scale models on extensive computational resources and has undergone comprehensive experimental validation.

2. UltraMem has significantly lower memory access cost during inference compared to MoE. Under common inference batch sizes, it can be up to **6 times** faster than MoE with the same parameters and calculations. The inference speed of UltraMem is almost identical to that of a dense model with equivalent computational resources.

3. We have verified the scaling ability of UltraMem. Similar to MoE, UltraMem has strong scaling ability, and we have observed stronger scaling ability than MoE.

## 2 RELATED WORK

**Mixture of Expert.** Shazeer et al. (2017) proposed MoE and Fedus et al. (2022) introduced the MoE in large language models, where each token selects one expert for inference each time, thereby increasing model parameters without increasing computation. Rajbhandari et al. (2022) introduced the concept of shared experts, where each token utilizes some fixed experts along with some unique experts. Subsequent research has focused on improving the gating functions of MoE, including token choice (Chi et al., 2022), non-trainable token choice (Roller et al., 2021) and expert choice (Zhou et al., 2022), primarily to address the issue of expert imbalance. Liu et al. (2024); Dai et al. (2024) opted to slice the experts into smaller segments while activating more experts per token, achieving significant performance improvements. Concurrent study (Krajewski et al., 2024) meticulously explored the benefits of granularity and increasing the number of experts, alongside investigating the scaling laws associated with MoE. In this paper, we use fine-grained MoE as our baseline, wherein the granularity of the MoE is set to 2. This means that each expert is half the size of the original MultiLayer Perceptron (MLP) , with two experts activated per token.

**Large Memory Layer.** Lample et al. (2019) first introduced the concept of large memory layer, called PKM, which can be seen as slicing the MoE experts to the smallest possible configuration. Kim & Jung (2020) introduced a concept similar to shared experts in MoE, allowing PKM and MLP to operate in parallel. Csordás et al. (2023) made a slight modification to PKM by removing the Softmax operation. PEER (He, 2024) improved the activation of values in PKM to activate a small expert with an inner dimension of 1, achieving significant performance gains. However, current research on PKM is limited to smaller models, and even the latest improved versions of PKM only outperform MoE in certain scenarios. Additionally, current PKM do not possess characteristics suitable for large-scale training. We address these issues in this paper.

**Tensor decomposition** breaks down a tensor into a series of small matrices or tensors. In deep learning research, such methods are commonly used to approximate a large tensor during training, aiming to save on computation and parameters. Product quantization (Jegou et al., 2010) breaks a vector into multiple sub-vectors, allowing us to reconstruct the original vector using a smaller number of sub-vectors, thereby reducing the model parameters. Bershatsky et al. (2024) initializes several matrices and a core tensor, trains these parameters during the fine-tuning phase, and reconstructs the original large tensor in a manner of Tucker Decomposition at the end of training to reduce training costs. We borrow this insight to improve PKM's key retrieval.

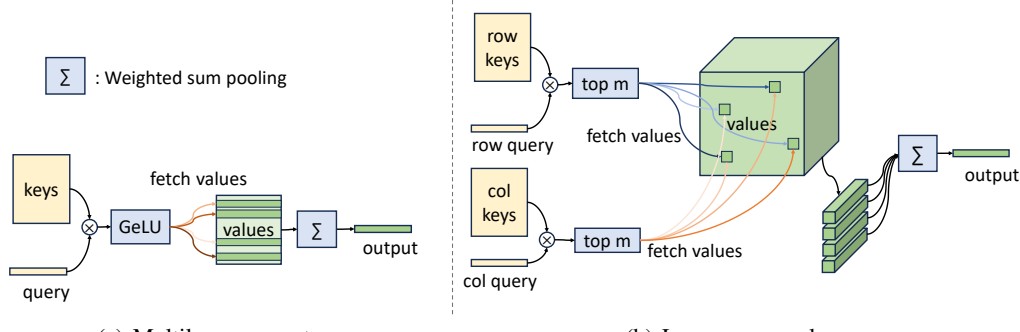

(a) Multilayer perceptron  (b) Large memory layer

Figure 2: An overview of multilayer perceptron (MLP) and large memory layer (LML). For the sake of brevity, we omit the third top-$m$ operation from memory layer. An MLP typically consists of two linear layers and a GeLU activation. We consider the weights of the first linear layer as keys, and those of the second linear layer as values. LML uses row and column keys to determine the 2-D logical address to index memory values, whereas MLP uses 1-D logical address. "fetch value" refers to retrieving values based on the indices with higher scores.

## 3 ULTRAMEM

### 3.1 PRELIMINARY

Here we firstly introduce the origin large memory layer (LML) based on product keys, which serves as the foundation for our proposed approach. The concept of a product key-based memory layer (PKM) was first explored in prior work (Lample et al., 2019). In their approach, the authors incorporated an external memory module into language models, with the goal of expanding the model's parameters while maintaining a similar level of computational complexity. The overall structural diagram is depicted in Figure 2(b).

A memory layer generally consists of two parts: keys $\mathbf{K} \in \mathbb{R}^{N \times D_k}$ and values $\mathbf{V} \in \mathbb{R}^{N \times D_v}$. To retrieve information from memory values, a query vector $\mathbf{q} \in \mathbb{R}^{D_k}$ finds most relevant values by multiplying keys to obtain scores. The higher the scores are, the better impact should the values have. Consequently, this process can be formulated as:

$$\mathbf{s} = \sigma(\mathbf{K}\mathbf{q}) \quad \mathbf{o} = \mathbf{V}^\top \mathbf{s}, \tag{1}$$

where $\mathbf{s}$ is the scores, $\sigma$ is a non-linear activation, $\mathbf{o}$ is the output. Attention layers, who memorize context contents, and MLP layers, who memorize world knowledge, also follow the above formulation with $\sigma$ being SoftMax in attention layers and GeLU in MLP layers (Geva et al., 2020) (see Figure 2(a)).

Product-key memory layers scale up the memory size with $N > 10^6$, while activating only a few values with top-$m$ scores. Here, $m$ is a hyper-parameter controlling sparsity. Though values are sparsely accessed, the keys, which are as large as values, must be fully computed to obtain scores before top-$m$ activation following equation 1. To alleviate the computation complexity for keys, product keys are proposed. Borrowing the idea of Product Quantization, it utilizes a 2-D logical address (see Figure 2(b)), typically a $n \times n$ grid where $n = \sqrt{N}$, for memory value retrieval. Specifically, a 2-D logical address $(i, j)$ is used to index memory value at physical address $n \times i + j$. With such strategy, logical scores are then represented as a matrix, which is further decomposed as an addition of row and column scores:

$$\mathbf{s}_{row} = \sigma_{\text{TopM}}(\mathbf{K}_{row} q_{row}(\mathbf{x})), \quad \mathbf{s}_{col} = \sigma_{\text{TopM}}(\mathbf{K}_{col} q_{col}(\mathbf{x})), \tag{2}$$

$$\mathbf{S}_{grid} = \sigma_{\text{TopM}}(\mathbf{s}_{row} + \mathbf{s}_{col}^\top), \quad \mathbf{o} = \mathbf{V}^\top \times \texttt{SoftMax}(\texttt{vec}(\mathbf{S}_{grid})), \tag{3}$$

where $\mathbf{K}_{row}, \mathbf{K}_{col} \in \mathbb{R}^{n \times D_k}$, $q_{row}, q_{col} : \mathbb{R}^{D_i} \to \mathbb{R}^{D_k}$ convert input hidden $\mathbf{x} \in \mathbb{R}^{D_i}$ to row and column query, $\sigma_{\text{TopM}}(\cdot)$ preserves top-$m$ largest elements in the input and set the rest to negative infinity, and the matrix addition with unmatched matrix shape is implemented by element broadcasting. It should be noted that removing $\sigma_{\text{TopM}}$ from equation 2 does not make any difference. The only

reason for applying top-$m$ to the row and column scores is to reduce the computation for the last top-$m$ operation on $\mathbf{S}_{grid}$. As $\mathbf{s}_{row}, \mathbf{s}_{col}$ have only $m$ activated scores, $\mathbf{S}_{grid}$ has only $m^2$ candidates for top-$m$ operation rather than $N$, i.e., top-$m$ complexity reduces from $O(N \log m)$ to $O((\sqrt{N} + m^2) \log m)$.

Note that the $\mathbf{S}_{grid}$ undergoes a SoftMax operation akin to the one employed in the self-attention mechanism. Moreover, PKM adopts the multi-head mechanism from the self-attention module, wherein it utilizes multiple key sets to retrieve the shared values, we denote $H$ as the number of PKM heads.

## 3.2 STRUCTURE IMPROVEMENTS

**Improve PKM with a bag of tricks.** We first studied the structure of PKM and found that a series of minor adjustments can steadily improve the model's performance:

1) We remove the operation Softmax in equation 3, which is well-established in the studies (Shen et al., 2023; Csordás et al., 2023).
2) We conduct Layer Normalization (LN) (Ba et al., 2016) on query and keys for stability of training.
3) PKM suggests using a constant learning rate of 0.001 to learn the values, which is much higher than the learning rate for other parameters. We found that gradually decaying the value learning rate provides further benefits.
4) PKM uses a linear layer to generate query, we add a causal depthwise convolutional layer (Howard, 2017) before this linear layer to enhance query.
5) Similar to Group Query Attention (Ainslie et al., 2023), we share query in two key sets. This can reduce the computational cost of generating the query by half, with little performance impact.
6) By halving $D_v$, we double the number of values. Under the condition of keeping the activation value parameter unchanged, we increased the diversity of activated values, and the model effect is further improved. In order to make the output consistent with hidden dimension, we add a linear layer on the aggregated output.

**UltraMem Overall structure.** We then take a deeper investigation into the model structure and propose UltraMem. Figure 3 shows the PKM and our improved UltraMem structure, based on a Pre-LayerNorm Transformer architecture. PKM replaces MLP or operates in parallel (Kim & Jung, 2020) with MLP in the one of deeper layers with memory layer. We notice three drawbacks to PKM:

1. As value size $N$ significantly increases, queries can harder find correct values.

2. Product key decomposition introduces bias on retrieval topology. For example, let $(i, j)$ be the logical address for the top-1 score, then top-2 score must be located on row $i$ or column $j$, which significantly limits the diversity of top-$m$ selection.

3. There are issues with unbalanced multi-GPU computation and communication during large-scale parameter training, as the full model parameters cannot be placed on a single GPU.

To alleviate problems 1 and 3, we decompose this large memory layer into multiple smaller memory layers distributed at fixed intervals across the transformer layers. Additionally, this skip-layer structure allows us to overlap the execution of the memory layer and the transformer layers, as the memory layer is predominantly memory-bound during training.

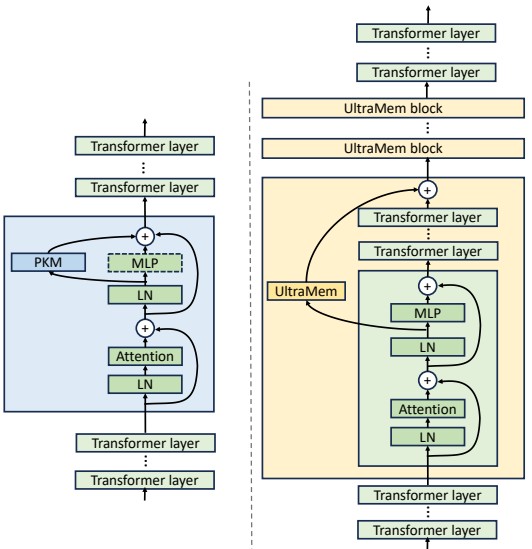

Figure 3: Overall of PKM and UltraMem.

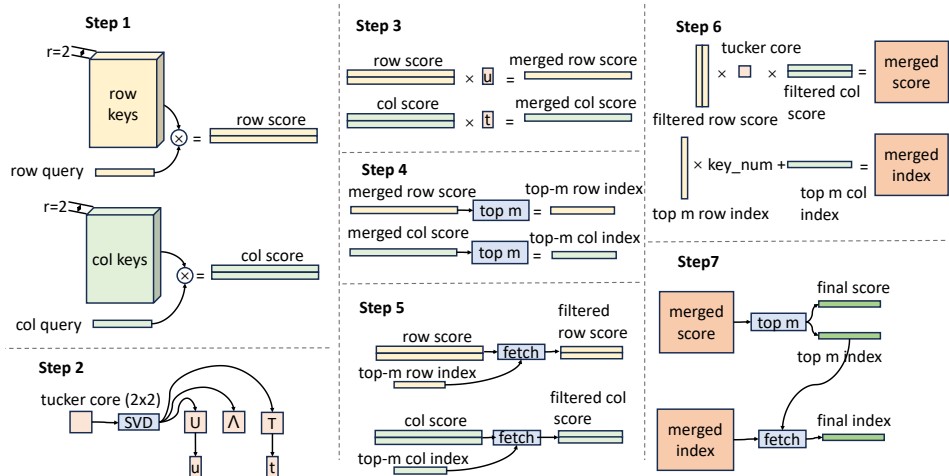

Figure 4: Flow of Tucker Decomposed Query-Key Retrieval (TDQKR), here $r = 2$. The term "fetch" refers to the action of retrieving scores based on a given index (corresponding to "torch.gather"). TDQKR replacing Product Quantization, serves as a more precise retrieval module for recalling value indices in UltraMem. Each step of the TDQKR process is meticulously referenced within the main text for understanding.

**Tucker Decomposed Query-Key Retrieval (TDQKR).** We explore a more complex multiplicative approach to alleviate problem 1 and 2, where a tucker decomposition (Malik & Becker, 2018) is adopted in place of product quantization. The whole process of TDQKR is illustrated in Figure 4. Specifically, tucker decomposition estimates grid scores with rank-$r$ matrix multiplication:

$$\mathbf{S}_{row} = \mathbf{K}_{row} q_{row}(\mathbf{x}), \qquad\qquad \mathbf{S}_{col} = \mathbf{K}_{col} q_{col}(\mathbf{x}), \qquad (4)$$

$$\mathbf{S}_{grid} = \sigma_{\text{TopM}}(\mathbf{S}_{row}^\top \times \mathbf{C} \times \mathbf{S}_{col}), \qquad\qquad (5)$$

where $\mathbf{S}_{row}, \mathbf{S}_{col} \in \mathbb{R}^{r \times n}$ and $C \in \mathbb{R}^{r \times r}$ is the tucker core, which is a learnable parameter with random initialization. To produce $n \times r$ shaped row and column score, the dimensions of the query and key are reshaped, resulting in $\mathbf{K}_{row}, \mathbf{K}_{col} \in \mathbb{R}^{r \times n \times (D_k/r)}$ and $q_{row}, q_{col} \in \mathbb{R}^{r \times (D_k/r)}$, corresponding to Figure 4 step 1.

However, equation 5 is inefficient to be directly applied in practice, as the top-$m$ operation cannot be simplified with an equivalent two-phase top-$m$ technique like product quantization can. As a consequence, we propose an approximated top-$m$ algorithm to tackle this problem. The key is to do rank-1 approximation for the tucker core, so that the overall top-$m$ can be approximated by:

$$\mathbf{C} \approx \mathbf{u}\mathbf{t}^\top, \quad \sigma_{\text{TopM}}(\mathbf{S}_{row}^\top \times \mathbf{C} \times \mathbf{S}_{col}) \approx \sigma_{\text{TopM}}((\mathbf{u}^\top \mathbf{S}_{row})^\top \times (\mathbf{t}^\top \mathbf{S}_{col})) \qquad (6)$$

where $\mathbf{u}, \mathbf{t} \in \mathbb{R}^{r \times 1}$. Note that $(\mathbf{u}^\top \mathbf{S}_{row}), (\mathbf{t}^\top \mathbf{S}_{col}) \in \mathbb{R}^{1 \times n}$ are row vectors, then the two-phase top-$m$ technique pertains to the approximated objective $\sigma_{\text{TopM}}((\mathbf{u}^\top \mathbf{S}_{row})^\top \times (\mathbf{t}^\top \mathbf{S}_{col}))$, corresponding to Figure 4 step 3. Overall, we conduct approximated top-$m$ on row and column scores, filtering out non-top elements, then we use the concrete objective in the final top-$m$ operated on $\mathbf{S}_{grid}$, keeping index scores precise:

$$\mathbf{C} \approx \mathbf{u}\mathbf{t}^\top \qquad (7) \qquad\qquad \tilde{\mathbf{S}}_{col} = \mathbb{I}_{\text{TopM}}(\mathbf{t}^\top \mathbf{S}_{col}) \odot \mathbf{S}_{col} \qquad (9)$$

$$\tilde{\mathbf{S}}_{row} = \mathbb{I}_{\text{TopM}}(\mathbf{u}^\top \mathbf{S}_{row}) \odot \mathbf{S}_{row} \qquad (8) \qquad\qquad \mathbf{S}_{grid} = \sigma_{\text{TopM}}(\tilde{\mathbf{S}}_{row}^\top \times \mathbf{C} \times \tilde{\mathbf{S}}_{col}), \qquad (10)$$

where $\mathbb{I}_{\text{TopM}}(\cdot)$ is binary value function, which converts top-$m$ elements to 1 and otherwise to 0. Equation 8&9 corresponding to Figure 4 step 4&5,and Equation 10 corresponding to Figure 4 step 6&7. As for the rank-1 approximation, we leverage Singular Value Decomposition (SVD) (Abdi, 2007) to factorize the tucker core with $\mathbf{u}, \mathbf{t}$ be the left and right singular vectors corresponding to the leading singular value, corresponding to Figure 4 step 2.

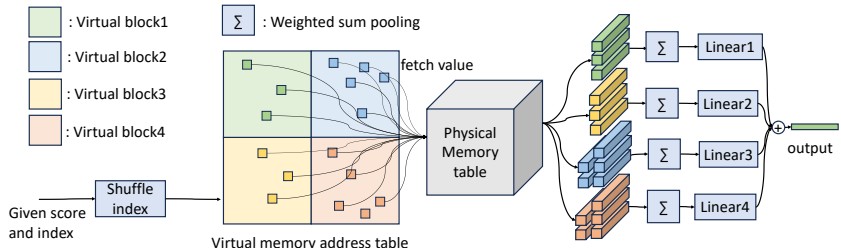

Figure 5: Flow of Implicit Value Expansion (IVE), here $E = 4$, $m = 16$. IVE reduces memory access and scales up memory size by expanding the memory table virtually. Each virtual block is a reparameterization of the physical memory table. Every virtual memory address corresponds to a physical memory address and a projector index. The weighted sum pooling is grouped by the virtual blocks, followed by a linear layer to produce the final output.

Last but not the least, the approximation error should be concerned when non-maximum singular values are as large as the maximum one. To mitigate this, an auxiliary loss that manages approximation error is introduced during training by constraining non-maximum eigenvalues:

$$\mathbf{C} = \mathbf{U}\Lambda\mathbf{T}^\top, \quad \text{(by SVD)} \tag{11}$$

$$\mathcal{L}_{aux} = \frac{\alpha}{r-1} \sum_{i=2}^{r} \left(\max\left(0, \lambda_i - \tau\right)\right)^2, \tag{12}$$

where, $\Lambda$ denotes the singular values for $\mathbf{C}$ in descending order, with $\tau$ serving as a margin to prevent $\mathbf{C}$ from degenerating into a rank-1 matrix, and $\alpha$ is the coefficient for the loss.

**Implicit Value Expansion (IVE).** Though sparsely used, maintaining a large memory table is still costly during training due to the large amount of memory access. To reduce memory access as well as scale up the memory size, we propose virtual memory as an implicit value expansion. Given a virtual expansion rate $E > 1$, virtual memory expands the memory table into $E$ times size. We design virtual memories as multiple reparameterizations of the original memory value $\mathbf{V}$, which we denote as physical memory. Then, $E$ linear projectors $\{\mathbf{W}_p | p \in [1, E], \mathbf{W}_p \in \mathbb{R}^{D_v \times D'_v}\}$ are utilized, and virtual memory block $\tilde{\mathbf{V}}_p$ corresponding to the $p$-th reparameterization can be defined as:

$$\tilde{\mathbf{V}}_p = \mathbf{V}\mathbf{W}_p. \tag{13}$$

Then the overall virtual memory is a concatenation of the virtual blocks $\tilde{\mathbf{V}} = [\tilde{\mathbf{V}}_0^\top, \tilde{\mathbf{V}}_1^\top, \ldots, \tilde{\mathbf{V}}_E^\top]^\top$. Note the dimension of virtual values $D'_v$ is not necessarily consistent with the dimension of physical values $D_v$.

To apply the virtual memory is intuitive, where memory table can be replaced from $\mathbf{V}$ to $\tilde{\mathbf{V}}$. And to fit virtual memory size, the key size is expanded by $\sqrt{E}$ times. Moreover, we suggest a random shuffle for virtual memory to eliminate some unnecessary index topology prior introduced by row and column scoring. Concretely, if the virtual memory tables are unioned by concatenation, each memory value and its expansions would be located in the same column in logical address, and thus can be potentially more frequently chosen simultaneously.

A naive reparameterization for virtual memory still introduces lots of computations, which is $E \cdot N \cdot D_v \cdot D'_v$, and $E$ times GPU memory access. A better idea is to compute reparameterization on demand. That is, we expand the logical address to triplets $(i, j, p)$ where $(i, j)$ is the original logical address and $p$ is index for the virtual memory block, and then simultaneously conduct sum pooling and compute virtual memory value. Consequently, equation 3 is rewritten as:

$$\hat{\mathbf{s}} = \texttt{Shuffle}(\texttt{vec}(\mathbf{S}_{grid})), \tag{14}$$

$$\mathbf{o} = \tilde{\mathbf{V}}^\top \times \hat{\mathbf{s}} = \sum_p \tilde{\mathbf{V}}_p^\top \times \hat{\mathbf{s}}_p = \sum_p \mathbf{W}_p^\top \left(\mathbf{V}^\top \times \hat{\mathbf{s}}_p\right) \tag{15}$$

where $\hat{\mathbf{s}}_p$ represents the scores corresponding to $p$-th virtual memory block. With equation 15, we can firstly lookup and pool values according to the virtual block index and then transform the reduced

physical values directly into reduced virtual values. This trick reduces extra computation from $E \cdot N \cdot D_v \cdot D_v'$ to $E \cdot B \cdot D_v \cdot D_v'$, where $B$ is the number of tokens in batch, and has nearly no extra GPU memory access except for the linear projectors. Figure 5 shows the flow of IVE.

**Multi-Core Scoring (MCS).** PKM shares a single score across dimension $D_v$ for each value. Empirically, assigning multiple scores to a single value has shown to enhance performance. Thus, we rewrite the tucker core $\mathbf{C}$ as a series of component cores $\mathbf{C} = \sum_i^h \mathbf{C}^{(i)}$. This allows employing $\{\mathbf{C}^{(i)}\}_{i=1}^h$ to generate individual score maps $\mathbf{S}_{tucker}^{(i)} = \mathbf{S}_{row}^\top \mathbf{C}^{(i)} \mathbf{S}_{col}$. Obviously,

$$\mathbf{S}_{tucker} = \mathbf{S}_{row}^\top (\sum_i^h \mathbf{C}^{(i)}) \mathbf{S}_{col} = \sum_i^h \mathbf{S}_{row}^\top \mathbf{C}^{(i)} \mathbf{S}_{col} = \sum_i^h \mathbf{S}_{tucker}^{(i)}. \qquad (16)$$

We keep top-$m$ conducted on aggregated score $\mathbf{S}_{tucker}$, while applying individual scores $\mathbf{S}_{tucker}^{(i)}$ on vertically split value table $\mathbf{V} = [\mathbf{V}^{(1)}, \ldots, \mathbf{V}^{(h)},]$ with $\mathbf{V}^{(i)} \in \mathbb{R}^{N \times (D_v/h)}$, i.e.,

$$\mathbf{o} = [\hat{\mathbf{s}}^{(1)\top} \mathbf{V}^{(1)}, \ldots, \hat{\mathbf{s}}^{(h)\top} \mathbf{V}^{(h)}]^\top. \qquad (17)$$

When this technique incorporates with IVE, we split physical memory values instead of virtual memory values to keep the equivalence in equation 15.

**Improved initialization.** PKM initializes values with a Gaussian distribution $\mathcal{N}(0, \frac{1}{D_v})$. Since PKM applies Softmax to the scores, the variance of the pooled outputs is $1/D_v$. We argue that LML should be considered as a component similar to an MLP and, therefore, should use an initialization method akin to that of MLPs. Before training, the output of an MLP typically follows a Gaussian distribution $\mathcal{N}(0, \frac{1}{2L})$ (Brown, 2020), where $L$ is the total number of layers. We initialize value with $\mathcal{N}(0, \frac{E}{2mHL})$, where $m$ is the activated value number, $H$ is the head number, $E$ is the value expansion times. To ensure that the output distribution of UltraMem is $\mathcal{N}(0, \frac{1}{2L})$, We need to confirm that the mean of top-$m$ score is 1, details see Appendix A.

## 4 QUANTITATIVE ANALYSIS WHY ULTRAMEM INSTEAD OF MOE

The most effective method for enhancing model capacity without significantly raising computational costs is MoE. This strategy employs a set of specialized sub-models, known as "experts", which work together to tackle complex problems. However, the MoE model poses challenges for inference processes.

Consider the Transformer hidden dimension as $D$, the inner dimension of MLP is $4D$, given the inference batch size as $B$. Using the MoE with $2\text{in}N_{moe}$ (choose 2 in $N_{moe}$ experts per token) as an example, where the inner dimension of expert is $2D$. Assuming the expert chosen is fully balanced, we can get the memory access of single MoE layer as $min(2B, N_{moe}) \times 2D^2$. For the UltraMem, assuming value dimension is $D/2$, and each token activates the top-$m$ values, then its memory access is $min(Bm, N) \times D/2$. As the batch size increases, the memory access of MoE grows rapidly until it reaches an upper limit where all expert parameters need to be accessed. In contrast, the memory access of UltraMem increases very slowly, only reaching parity with MoE when the batch size is in the tens of thousands. However, in inference scenarios, the batch size is typically not very large.

Figure 1 shows the inference time and memory access of a 1.6 billion parameter Transformer with 2in34 MoE and $\times 12$[1] UltraMem. For larger batch sizes, see Figure 7 in Appendix. Compared to MoE, UltraMem achieves the maximum acceleration of $\times 6$ at a batch size of 64, and also shows significant acceleration at other batch sizes.

## 5 EXPERIMENTS

In this section, we demonstrate the scaling capabilities of UltraMem, showing that it outperforms MoE. We additionally show how the performance of UltraMem varies with different top-$m$ values and the number of parameters, and perform an ablation study to measure the impact of each part of UltraMem.

---

[1]The number of parameters in UltraMem is 12 times the number of parameters in the dense layer. In this case, the total parameters and total computation of UltraMem are the same as the 2in34 MoE.

## 5.1 SETUP

**Datasets.** Training data comes from RedPajama (Computer, 2023), containing 1 trillion tokens. RedPajama represents a clean-room, fully open-source version of the LLaMa (Touvron et al., 2023) dataset. Validation data includes the C4 validation set (Raffel et al., 2020), derived from the Common Crawl web corpus. The C4 training set is also incorporated within the RedPajama training data.

**Tokenizer** is based on the GPT-NeoX (Black et al., 2022) tokenizer, which uses the Byte-Pair Encoding (BPE) (Sennrich et al., 2015) algorithm and has a vocabulary size of 50,432.

**Evaluation.** We conducted a comprehensive evaluation of all models across ten benchmark datasets. These datasets included MMLU, Trivia-QA, GPQA, and ARC for assessing the models' **knowledge** capabilities; BBH, BoolQ, HellaSwag, and WinoGrande for evaluating **reasoning** skills; DROP for testing **reading comprehension abilities**; and AGIeval for measuring overall model performance. The decoding hyperparameters are aligned with those of LLaMA3 (Dubey et al., 2024). Detrails see Appendix E.

**Training details.** We used a standard pre-norm transformer (Xiong et al., 2020) with rotary embeddings (Su et al., 2024) for our dense models, which have 151M, 680M, 1.6B, and 6.5B parameters[2]. For sparse models, including UltraMem, PKM and MoE, we expand the sparse parameters twelvefold from the 151M, 680M, and 1.6B dense models. In MoE models, two experts are activated per token (Jiang et al., 2024), using a balance loss (Fedus et al., 2022) weight of 0.01 to ensure even expert selection. We slightly increased the width of MoE's experts to match UltraMem's computational and parameter costs. In UltraMem models, the auxiliary loss weight is $\alpha = 0.001$ and margin $\tau = 0.15$. The learning rate for values is ten times other parameters and decays linearly. For model structure and hyperparameters details, see Appendix E, and for large-scale training optimizations, see Appendix C, D.

## 5.2 EVALUATION ON LANGUAGE MODELING DATASETS

We evaluate models of various sizes, the results are shown in Table 1[3], where FLOPs is the computation cost of single token, the curves showing changes over the course of training are provided in the Figure 11 in Appendix. We observe that as the model capacity increases, UltraMem can outperform PKM and MoE with the same parameter and computation. On the 1.6B dense model, an UltraMem model with 12x the parameters can match the performance of a 6.5B dense model.

Table 1: Performance metrics of various models.

| Model | Param (B) | FLOPs (G) | Val. loss↓ | GPQA↑ | TriviaQA↑ | BBH cot↑ | Hella Swag↑ | Wino Grande↑ | DROP↑ | Avg↑ |
|---|---|---|---|---|---|---|---|---|---|---|
| Dense-151M | 0.15 | 0.30 | 2.96 | 19.98 | 12.67 | 22.57 | 35.07 | 52.49 | 13.60 | 26.06 |
| PKM-151M-x12 | 2.04 | 0.35 | 2.76 | 17.30 | 24.66 | 23.14 | 42.25 | 51.38 | 13.10 | 28.64 |
| MoE-151M-2in32 | 2.04 | 0.35 | 2.63 | 17.30 | 33.27 | 23.24 | 48.44 | 55.96 | 18.57 | **33.20** |
| UltraMem-151M-x12 | 2.03 | 0.35 | 2.67 | 19.42 | 28.97 | 22.65 | 43.96 | 50.83 | 14.08 | 29.99 |
| Dense-680M | 0.68 | 1.36 | 2.64 | 21.09 | 27.16 | 24.65 | 48.83 | 54.93 | 22.97 | 33.27 |
| PKM-680M-x12 | 8.95 | 1.50 | 2.46 | 20.65 | 46.31 | 26.97 | 57.32 | 61.72 | 25.20 | 39.70 |
| MoE-680M-2in33 | 8.95 | 1.50 | 2.39 | 20.54 | 34.19 | 26.63 | 62.71 | 59.98 | 26.54 | 38.43 |
| UltraMem-680M-x12 | 8.93 | 1.49 | 2.37 | 21.99 | 55.17 | 26.62 | 64.15 | 60.54 | 25.14 | **42.27** |
| Dense-1.6B | 1.61 | 3.21 | 2.49 | 21.76 | 39.65 | 26.41 | 58.6 | 61.72 | 22.63 | 38.46 |
| PKM-1.6B-x12 | 21.13 | 3.48 | 2.34 | 22.99 | 48.92 | 28.98 | 65.45 | 63.93 | 27.55 | 42.97 |
| MoE-1.6B-2in34 | 21.36 | 3.52 | 2.30 | 21.32 | 59.56 | 29.46 | 67.34 | 63.93 | 28.81 | 45.07 |
| UltraMem-1.6B-x12 | 21.41 | 3.50 | 2.24 | 24.66 | 66.38 | 30.63 | 71.52 | 66.38 | 29.99 | **48.26** |
| Dense-6.5B | 6.44 | 12.88 | 2.30 | 19.98 | 57.28 | 31.14 | 69.73 | 65.9 | 33.12 | 46.19 |

---

[2]Excludes tokenizer vocabulary embedding and prediction head parameters.

[3]This table only includes evaluation results where the metrics have steadily increased with training. For all results, see the Table 7 in Appendix.

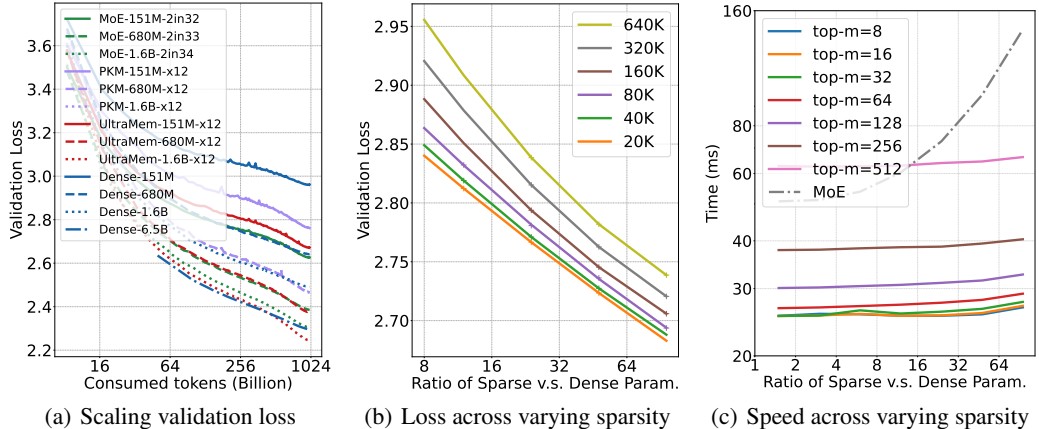

| (a) Scaling validation loss | (b) Loss across varying sparsity | (c) Speed across varying sparsity |

Figure 6: (a). C4 validation loss of different models at different scale. (b). Scaling curves at different sparsity with 151M activated parameters. Each line represents the same model sparsity; e.g., 20K indicates that approximately one out of every 20,000 values will be activated. The loss decreases linearly as the sparse parameters increase exponentially. (c). Inference time for UltraMem and MoE with 1.6B activated parameters. The batch size is 512, sequence length is 1, and key/value cache length is 2048. With fixed activation parameters, UltraMem's inference time remains nearly constant as sparse parameters increase, while MoE's inference time increases significantly.

## 5.3 VALUE NUMBER AND TOP-$m$

In most sparse LLMs, such as MoE and UltraMem, there is a clear positive correlation between sparsity and model performance. Therefore, in this section, we conduct a series of scaling experiments by varying selected top-$m$ and value number, i.e., the parameters of the sparse modules, to verify the changes in model performance with respect to sparsity. The result is shown in Figure 6(b).

It is evident that, at the same level of sparsity, the validation loss decreases as the number of parameters increases and can maintain a certain degree of decline. Additionally, the smaller the sparsity, i.e., the larger the proportion of activated parameters, the better the model performance. However, this also results in a higher memory access overhead. Thus, there is a trade-off between memory access volume and scaling efficiency. In our final experiments, we selected a sparsity ratio of 80K as the default model configuration.

As sparse parameters increase, Figure 6(c) shows that UltraMem maintains stable inference time despite exponential growth in parameters, as long as activated parameters (top-m) stay constant. In contrast, MoE's inference time rises significantly under similar conditions. Additionally, Figure 1(b) demonstrates that with smaller batch sizes, MoE's inference speed deteriorates even more compared to UltraMem.

## 5.4 ABLATION

We conduct comprehensive ablation studies based on the 151M dense model. In the baseline, the PKM is a version that operates in parallel with the MLP, making it a stronger baseline. For this group of experiments, the learning rate (LR) is set to 1.2e-4, with training on 500B tokens and evaluating the cross entropy loss on the training and C4 validation sets. We ensure that the parameter count and computational cost of the final version of the model were essentially at the same level.

Table 2 shows the ablation results. We identify 6 changes that significantly improved performance:

1. Doubling the number of values while halving their dimension, and simultaneously double the top-$m$ selections to keep the active parameters consistent.

2. Splitting a single UltraMem into multiple smaller units evenly across the transformer layers, with outputs skipping several blocks. This arrangement keeps the total parameter count, computational cost, and sparse parameter activation at or below pre-split levels.

Table 2: Ablation study of model improvements

| | Train Loss ↓ | Valid. Loss ↓ | Dense Param.(M) | Sparse Param.(G) | FLOPs (M) |
|---|---|---|---|---|---|
| PKM-151M-x10 | 2.604 | 2.828 | 173.01 | 1.534 | 346.06 |
| +rm softmax | 2.570 -0.034 | 2.822 -0.006 | 173.01 | 1.534 | 346.06 |
| +half vdim+proj | 2.556 -0.014 | 2.800 -0.022 | 178.47 | 1.529 | 356.98 |
| +share query | 2.560 +0.004 | 2.803 +0.003 | 173.46 | 1.529 | 346.96 |
| +split big mem&skip | 2.554 -0.006 | 2.788 -0.015 | 161.64 | 1.536 | 323.32 |
| +query/key LN | 2.553 -0.001 | 2.789 +0.001 | 161.64 | 1.536 | 323.54 |
| +IVE | 2.544 -0.009 | 2.772 -0.017 | 172.37 | 1.536 | 344.98 |
| +TDQKR | 2.538 -0.006 | 2.764 -0.008 | 172.37 | 1.536 | 344.98 |
| +MCS | 2.521 -0.017 | 2.761 -0.003 | 172.37 | 1.536 | 344.98 |
| +improved init | 2.518 -0.003 | 2.758 -0.003 | 172.37 | 1.536 | 344.98 |
| +value lr decay | 2.494 -0.024 | 2.736 -0.022 | 172.37 | 1.536 | 344.98 |
| +query conv | 2.493 -0.001 | 2.736 -0.000 | 172.38 | 1.536 | 345.02 |
| **Total Diff** | **-0.111** | **-0.092** | **-0.64** | **+0.002** | **-1.04** |

3. Tucker Decomposition Query-Key Retrieval introduces negligible additional parameters while reducing computation, here $r = 2$.

4. Multi-Core Scoring significantly reduces training loss, and slightly reduces validation loss, here $h = 2$.

5. Implicit Value Expansion slightly increases both the parameter count and computational cost, but the improvement is significant, here $E = 4$.

6. The LR for the value parameters starts at ten times that of the other parameters and linearly decays to match them by the end of training.

Among other changes, sharing the query helps cut computational costs with a minor trade-off in performance. Normalizing the query/key greatly reduces spikes in training perplexity and enhances training stability, as shown in Figure 10.(a). Improved initialization prevents score and output variance explosions in the early to middle training stages, detailed in Figure 10.(b) and (c). Additionally, employing convolution further limits the variance divergence in UltraMem outputs(Figure 10.(c)). The above results are based on incremental ablation. Results from independent ablation can be found in the Table 8 in Appendix, and they align with our expectations.

Beside, We conduct another ablation studies on IVE, TDQKR, and MCS with different configurations, which are documented in Table 3. For IVE, as $E$ increases, there is a consistent improvement in model performance alongside a notable increase in computational cost. However, the marginal gains decrease as $E$ rises, leading us to recommend $E = 4$. For TDQKR and MCS, increasing $r$ and $h$ does not significantly change the computational load, but the effectiveness no longer shows marked improvement, hence we suggest using $r = 2$ and $h = 2$.

Table 3: Ablation of different config on IVE, TDQKR, and MCS

| | IVE | | | | TDQKR | | | | MCS | | | |
|---|---|---|---|---|---|---|---|---|---|---|---|---|
| | Baseline | E=4 | E=9 | E=16 | Baseline | r=2 | r=3 | r=4 | Baseline | h=2 | h=4 | h=8 |
| Training loss↓ | 2.553 | -0.009 | -0.016 | -0.019 | 2.544 | -0.006 | -0.0065 | -0.0063 | 2.538 | -0.017 | -0.017 | -0.012 |
| Validation loss↓ | 2.789 | -0.017 | -0.025 | -0.027 | 2.772 | -0.008 | -0.0084 | -0.0082 | 2.764 | -0.003 | +0.001 | +0.006 |
| FLOPs(G) | 323.54 | +6.6% | +14.9% | +26.4% | 344.98 | +0.001% | +0.002% | +0.003% | 344.98 | +0.001% | +0.003% | +0.007% |

## 6 CONCLUSION

In this paper, we introduce UltraMem, which, compared to MoE, has minimal memory access and therefore achieves up to a **sixfold** speed advantage. Concurrently, in terms of performance, UltraMem **surpasses** MoE with the same parameters and computation as model capacity increases, indicating its superior scaling capability. This work presents a promising direction for developing more efficient and scalable language models.

ACKNOWLEDGMENTS

We extend our deepest gratitude to Pingshuo Ma and Wenda Liu for their invaluable assistance in optimizing the early stage training of large-scale UltraMem. We also appreciate the inference optimization work for the UltraMem carried out by Siyan Chen, as well as Fan Xia's efforts in assessing the inference speed of MoE.

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

# A    ULTRAMEM INITIALIZATION

We initialize value with $\mathcal{N}(0, \frac{E}{2kHL})$, where $k$ is the activated value number, $H$ is the head number, $E$ is the value expansion times. To ensure that the output distribution of UltraMem is $\mathcal{N}(0, \frac{1}{2L})$, We need to confirm that the mean of top-$m$ score is 1.

Assuming the candidate score follows $\mathcal{N}(0, 1)$, and $k \ll \mathcal{K}$. We can simplify the problem as follows: Given N standard Gaussian distributed random variables $X_1, ..., X_n$, and the random variable $Y = mean(topm(X_1, ..., X_n))$, find the expected value $E(Y)$. It is difficult to obtain an analytical solution for E(Y), so we approximate E(Y) by sampling M times N points from a Gaussian distribution and calculating the mean of the top-$m$ values.

Then we initialize the query layer norm weight as $1/\sqrt{E(Y)}$, the keys layer norm weight as $1/\sqrt{D_k}$ to ensure the expected of candidate score is 1.

# B    INFERENCE TIME AND MEMORY ACCESS

Figure 7 shows that UltraMem has a much slower growth in memory access compared to MoE, only aligning with MoE in terms of memory access when the batch size reaches 131,072, and it continues to have an advantage in inference speed.

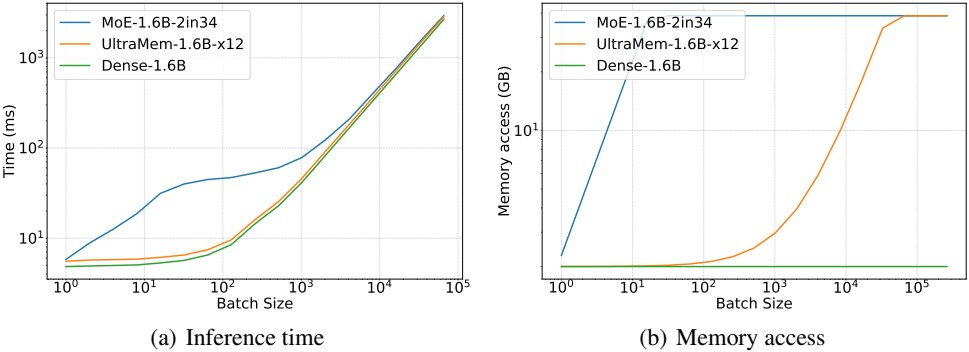

(a) Inference time          (b) Memory access

Figure 7: Inference time and memory access of Transformer, MoE and UltraMem. We ensured that three models have the same computation, and MoE and UltraMem have the same parameters. The x-axis and y-axis are both plotted on a logarithmic scale. The sequence length is 1 because during inference, we can only predict one token at a time, and the key/value cache length is 2048. The modes run on the A100-SXM.

# C    MEGATRON SUPPORT FOR TRAINING EFFICIENCY

As memory table scales towards billions even trillions of parameters, model parallelism becomes essential to distribute model parameters and optimizer states across multiple devices to ensure they fit into device memory and are trainable within a reasonable time frame. We leverages Megatron's (Shoeybi et al., 2019; Narayanan et al., 2021) 3D parallelism (pipeline parallelism, data parallelism, and tensor parallelism) for training. However, several parallelism modifications are required to support the memory table effectively. Because pipeline parallelism cannot address scenarios where a single layer's parameters exceed the memory capacity of a single device, and tensor parallelism is typically limited to a relatively small group of GPUs, making it insufficient to meet the memory table's memory requirements. Consequently, we propose sharding the memory table across a combination of data parallel and tensor parallel groups or its subgroups, to ensure efficient distribution and scalability.

The memory table can be partitioned either number-wise or dimension-wise. The entire process of number-wise and dimension-wise partitioning, along with their communication volume analysis and guidance on how to choose the appropriate partitioning method, is detailed in Appendix D. In our structural improvements, halving v_dim can simultaneously reduce the communication overhead for

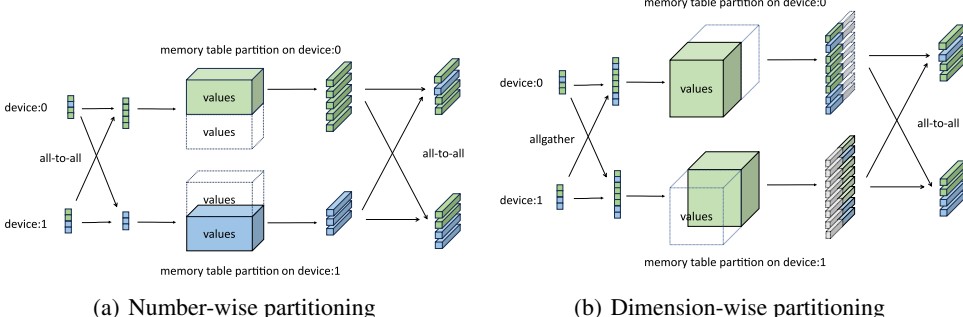

(a) Number-wise partitioning            (b) Dimension-wise partitioning

Figure 8: Process of Number-wise partitioning and Dimension-wise-partitioning. The weighted sum pooling step is omitted in the diagram.

both number-wise and dimension-wise partitioning. However, increasing top-$m$ will proportionally increase the communication overhead. Additionally, Implicit Value Expansion, due to the increase in the size of values after weighted sum pooling, will further impact the communication volume for dimension-wise partitioning.

To further augment performance, several key modifications have been implemented:

**Fused Lookup-Reduce Operator**: This newly introduced operator accelerates computations and reduces memory usage by combining the lookup and weighted sum pooling operations into a single, more efficient step.

**Asynchronous Execution Strategy**: Recognizing the benefits of cross-layer utilization of the memory layer, we have adopted an asynchronous execution strategy. This strategic choice allows for the concurrent processing of memory calculations alongside dense network operations, substantially enhancing the overall system performance.

These enhancements demonstrate the efficacy of our parallelism strategy within the Megatron framework, paving the way for more efficient training of large-scale models.

## D  NUMBER-WISE AND DIMENSION-WISE PARTITION DETAILS

Figure 8 shows the process of number-wise and dimension-wise partition. For number-wise partitioning, we first perform an all-to-all on indices to distribute them to the corresponding devices. After the lookup operation, the results are sent back to the original devices, then do weighted sum pooling. For dimension-wise partitioning, we need to perform an all-gather operation on indices to obtain all indices across devices. The lookup operation is then performed, dimension-wise partitioning allows the results to be sent back to each device after completing the weighted sum pooling.

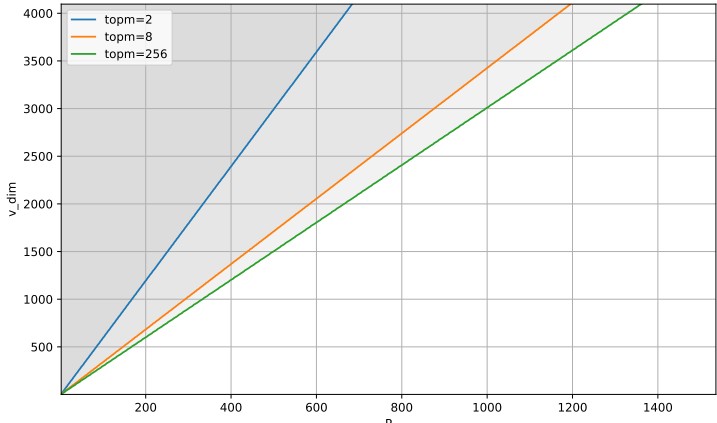

Figure 9: Relationship between P and v_dim for communication volume of number-wise / dimension-wise equals 1, the shaded area is number-wise / dimension-wise greater than 1

Assuming the memory table is distributed across P processors, the communication volume can be described as follows:

Number-wise Partitioning Communication Volume (not considering indices deduplication):

- All-to-all transmission of indices: $sizeof(int) \times bs \times topm \times (P-1)/P$

- All-to-all transmission of embeddings after lookup: $sizeof(bfloat16) \times bs \times topm \times v\_dim \times (P-1)/P$

Dimension-wise Partitioning Communication Volume:

- AllGather indices: $sizeof(int) \times bs \times topm \times (P-1)$

- AllGather scores: $sizeof(bfloat16) \times bs \times topm \times (P-1)$

- All-to-all transmission of embeddings post-lookup reduction: $sizeof(bfloat16) \times bs \times v\_dim \times (P-1)/P$

Here $v\_dim$ is the value dimension, $bs$ is the batch size times sequence length. Figure 9 shows the relationship between P and v_dim for communication volume of these two partitioning methods, helping us choose the appropriate partitioning method under a fixed configuration.

# E  EXPERIMENT SETTING

Table 4 displays common hyper-parameter settings for all experiments. "LR" stands for Learning Rate, corresponding to the values 6e-4, 2.5e-4, 2e-4, and 1.2e-4 for dense models with sizes 151M, 680M, 1.6B, and 6.5B, respectively (Brown, 2020). Regarding the insertion of UltraMem and PKM, for UltraMem-151M, it's 3:5/6:8/9:11, where 3:5 indicates that UltraMem input is taken from layer 3 and inserted back into the output of layer 5, and so on. For UltraMem-680M, it's 3:7/8:12/13:17/18:22. For UltraMem-1.6B, it's 3:7/8:12/13:17/18:22/23:27/28:32. For PKM-151M, it's 6:6. For PKM-680M, it's 12:12. For PKM-1.6B, it's 16:16. The settings for UltraMem and MoE models align with their dense counterparts based on dense parameter size. Table 6 shows the model parameter setting used in scaling experiments. What's more, the common setting for UltraMem is shown in Table 5.

| Configuration Key | Value |
|---|---|
| Weight decay | 0.1 |
| $\beta_1$ | 0.9 |
| $\beta_2$ | 0.95 |
| LR | 6e-4/2.5e-4/2e-4/1.2e-4 |
| LR end ratio | 0.1 |
| LR schedule | cosine |
| LR warmup ratio | 0.01 |
| Dropout | 0.1 |
| Batch size | 2048 |
| Sequence length | 2048 |
| Training step | 238418 |

Table 4: Training hyper-parameters

| Configuration Key | Value |
|---|---|
| Tucker rank $r$ | 2 |
| Multi-core scoring $h$ | 2 |
| Virtual memory expansion $E$ | 4 |
| Aux loss weight $\alpha$ | 0.001 |
| Aux loss margin $\tau$ | 0.15 |

Table 5: Common UltraMem configuration

**Evaludation datasets.** We use 10 benchmarks to evaluate all kind of models.

1. Knowledge: Massive Multitask Language Understanding (MMLU) (Hendrycks et al., 2020), TriviaQA (Joshi et al., 2017), Graduate-Level Google-Proof Q&A Benchmark (GPQA) (Rein et al., 2023), AI2 Reasoning Challenge (ARC) (Clark et al., 2018).

2. Reasoning: BIG-Bench Hard (BBH) (Suzgun et al., 2022), Boolean Questions (BoolQ) (Clark et al., 2019), HellaSwag (Hella) (Zellers et al., 2019), WinoGrande (Wino) (Sakaguchi et al., 2021).

3. Reading comprehension: Discrete Reasoning Over Paragraphs (DROP) (Dua et al., 2019).

4. Comprehensive ability: AGIEval (Zhong et al., 2023)

| Model | Hidden Dim | Inner Dim | Attn Head | Layer | Top-m | Expert | Kdim | Knum | Mem Layer | Param (B) | FLOPs (G) |
|---|---|---|---|---|---|---|---|---|---|---|---|
| Dense-151M | 1024 | 4096 | 16 | 12 | - | - | - | - | - | 0.15 | 0.30 |
| Dense-680M | 1536 | 6144 | 16 | 24 | - | - | - | - | - | 0.68 | 1.36 |
| Dense-1.6B | 2048 | 8192 | 16 | 32 | - | - | - | - | - | 1.61 | 3.21 |
| Dense-6.5B | 4096 | 16384 | 32 | 32 | - | - | - | - | - | 6.44 | 12.88 |
| MoE-151M-2in32 | 1024 | 2528 | 16 | 12 | 2 | 32 | - | - | - | 2.04 | 0.35 |
| MoE-680M-2in33 | 1536 | 3584 | 16 | 24 | 2 | 33 | - | - | - | 8.95 | 1.50 |
| MoE-1.6B-2in34 | 2048 | 4672 | 16 | 32 | 2 | 34 | - | - | - | 21.36 | 3.52 |
| PKM-151M-x12 | 1024 | 4096 | 16 | 12 | 16x6 | - | 512 | 1347 | 1 | 2.04 | 0.35 |
| PKM-680M-x12 | 1536 | 6144 | 16 | 24 | 35x8 | - | 768 | 2308 | 1 | 8.95 | 1.50 |
| PKM-1.6B-x12 | 2048 | 8192 | 16 | 32 | 42x12 | - | 896 | 1792 | 1 | 21.44 | 3.52 |
| UltraMem-151M-x10 | 1024 | 4096 | 16 | 12 | 16x2 | - | 256 | 1024 | 3 | 1.71 | 0.35 |
| UltraMem-151M-x12 | 1024 | 4096 | 16 | 12 | 16x2 | - | 256 | 1100 | 3 | 2.03 | 0.35 |
| UltraMem-680M-x12 | 1536 | 6144 | 16 | 24 | 35x2 | - | 384 | 1632 | 4 | 8.93 | 1.49 |
| UltraMem-1.6B-x12 | 2048 | 8192 | 16 | 32 | 42x2 | - | 448 | 1792 | 6 | 21.41 | 3.50 |

Table 6: Model parameter setting. Top-$m$ means chosen expert number in MoE, means chosen value number times head number in PKM and UltraMem. Kdim means the key dimension in PKM and UltraMem. Knum means the number of keys, Knum$^2$ is the number of values.

# F  MORE EXPERIMENT RESULTS

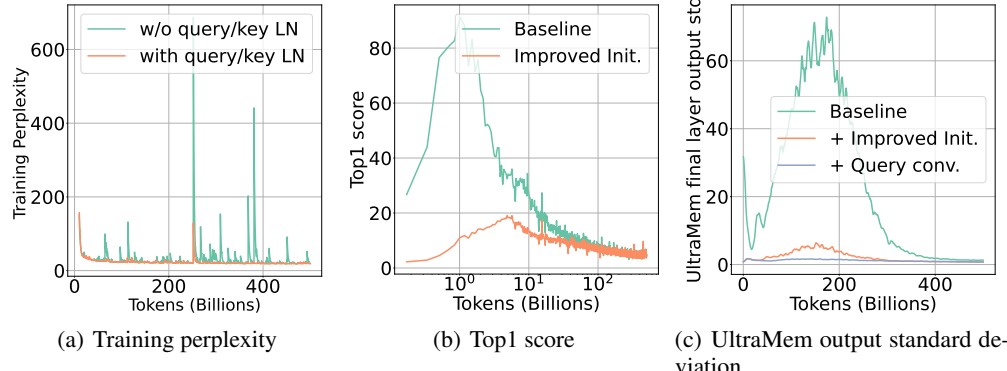

(a) Training perplexity  (b) Top1 score  (c) UltraMem output standard deviation

Figure 10: Model training state details. "Top1 Score" refers to the highest score among the retrieved keys. "UltraMem Output Std" represents the standard deviation of the outputs from the last layer of UltraMem.

Table 7: All performance metrics of various models

| Model | Param (B) | FLOPs (G) | ARC-C↑ | GPQA↑ | Trivia QA↑ | MMLU↑ | BBH cot↑ | BoolQ↑ | Hella Swag↑ | Wino Grande↑ | AGI Eval↑ | DROP↑ | Avg↑ |
|---|---|---|---|---|---|---|---|---|---|---|---|---|---|
| Dense-151M | 0.15 | 0.30 | 25.60 | 19.98 | 12.67 | 26.50 | 22.57 | 50.15 | 35.07 | 52.49 | 9.03 | 13.60 | 26.77 |
| PKM-151M-x12 | 2.04 | 0.35 | 25.94 | 17.30 | 24.66 | 25.69 | 23.14 | 53.48 | 42.25 | 51.38 | 9.65 | 13.10 | 28.66 |
| MoE-151M-2in32 | 2.04 | 0.35 | 26.96 | 17.30 | 33.27 | 26.58 | 23.24 | 55.96 | 48.44 | 55.96 | 9.34 | 18.57 | **31.56** |
| UltraMem-151M-x12 | 2.03 | 0.35 | 25.68 | 19.42 | 28.97 | 25.62 | 22.65 | 47.74 | 43.96 | 50.83 | 10.00 | 14.08 | 28.89 |
| Dense-680M | 0.68 | 1.36 | 24.06 | 21.09 | 27.16 | 24.64 | 24.65 | 46.42 | 48.83 | 54.93 | 9.44 | 22.97 | 30.42 |
| PKM-680M-x12 | 8.95 | 1.50 | 25.51 | 20.65 | 46.31 | 25.22 | 26.98 | 41.80 | 57.32 | 61.72 | 8.94 | 25.20 | 33.97 |
| MoE-680M-2in33 | 8.95 | 1.50 | 25.17 | 20.54 | 34.19 | 24.38 | 26.63 | 43.70 | 62.71 | 59.98 | 7.39 | 26.54 | 33.13 |
| UltraMem-680M-x12 | 8.93 | 1.49 | 23.72 | 21.99 | 55.17 | 24.97 | 26.62 | 48.20 | 64.15 | 60.54 | 8.26 | 25.14 | **35.88** |
| Dense-1.6B | 1.61 | 3.21 | 26.30 | 21.76 | 39.65 | 26.19 | 26.41 | 51.50 | 58.6 | 61.72 | 9.22 | 22.63 | 34.81 |
| PKM-1.6B-x12 | 21.13 | 3.48 | 26.71 | 22.99 | 48.92 | 24.80 | 28.98 | 60.06 | 65.46 | 63.93 | 9.51 | 27.55 | 37.89 |
| MoE-1.6B-2in34 | 21.36 | 3.52 | 25.43 | 21.32 | 59.56 | 26.18 | 29.46 | 42.78 | 67.34 | 63.93 | 6.63 | 28.81 | 37.14 |
| UltraMem-1.6B-x12 | 21.41 | 3.50 | 25.94 | 24.66 | 66.38 | 24.67 | 30.63 | 59.8 | 71.52 | 66.38 | 8.77 | 29.99 | **40.88** |
| Dense-6.5B | 6.44 | 12.88 | 28.16 | 19.98 | 57.28 | 27.68 | 31.14 | 68.2 | 69.73 | 65.9 | 9.23 | 33.12 | 41.04 |

Table 8: Independent ablation study of model improvements

|  | Train Loss ↓ | Valid. Loss ↓ |
|---|---|---|
| PKM-151M-x10 | 2.604 | 2.828 |
| + rm softmax | -0.034 | -0.006 |
| + half vdim+proj | -0.027 | -0.02 |
| + share query | -0.003 | -0.002 |
| + split big mem | -0.003 | -0.005 |
| + query/key LN | -0.002 | +0.003 |
| + IVE | -0.025 | -0.023 |
| + TDQKR | -0.003 | -0.007 |
| + TDQKR + MCS | -0.02 | -0.009 |
| + value lr decay | -0.017 | -0.007 |
| + query conv | -0.005 | -0.001 |

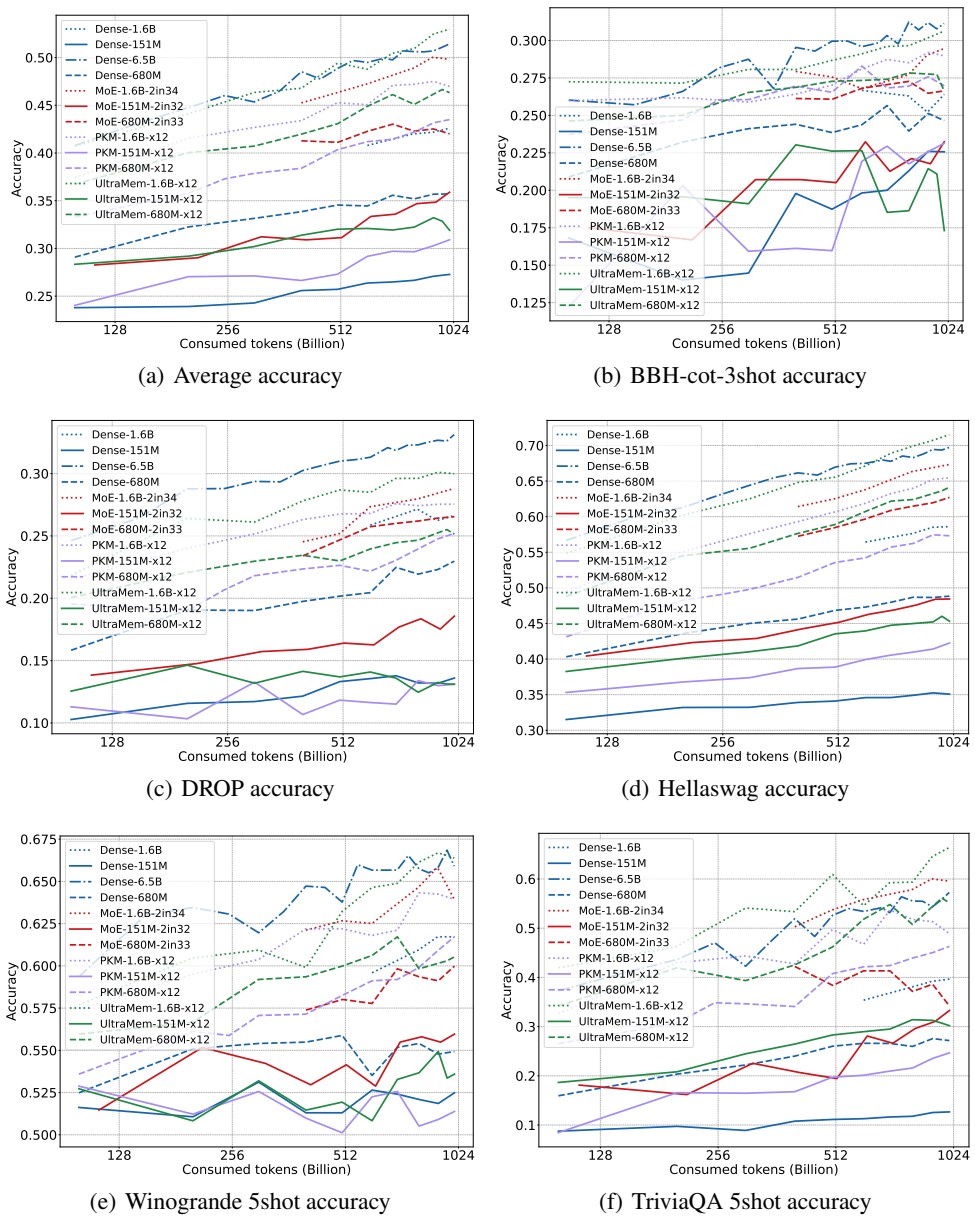

Figure 11: The changes in accuracy for all observable evaluation throughout the training.

