# OpenReview forum: "Ultra-Sparse Memory Network"
_ICLR.cc/2025/Conference — ICLR 2025 Poster_

### Official Review · Reviewer_bPzv · 2024-10-28

**Soundness:** 3
**Presentation:** 2
**Contribution:** 3
**Rating:** 6
**Confidence:** 2

**Summary:**

This paper introduces a novel layer designed to increase model size efficiently. Specifically, the approach refines the existing PKM method and incorporates TDQKR as a sparse method to enhance performance. Additionally, it proposes implicit value expansion to address memory access challenges in large models. A comparative analysis of memory access between the proposed method and MOE is also provided. Experimental results demonstrate significant performance improvements.

**Strengths:**

1. The techniques proposed in this paper are novel. On one hand, the paper thoroughly studies the techniques in PKM, offering valuable empirical insights for future research. On the other hand, it integrates algorithms with memory management to address the memory access issue.
2. The paper includes an efficiency analysis focused on memory access.
3. Experimental results show that the method significantly outperforms existing approaches on large models, being up to six times faster than MOE at the same scale.

**Weaknesses:**

1. The writing could be improved, particularly since it introduces concepts beyond the domain of algorithms. These should be better explained as background information. For example, the implicit value expansion technique is somewhat confusing when the paper discusses virtual and physical memory, and could benefit from further clarification.

2. The experimental results in the main paper do not fully support the claims made in the methods section. First, while the paper asserts that the method improves PKM, there is no direct comparison except for validation loss. Second, although the paper claims that its memory access method outperforms MOE, this is only demonstrated in the appendix. The main table merely shows FLOPs and model parameters, which do not sufficiently illustrate the method's advantages (including the running time).

### Minor:
1. The dimensions in Equation (6) are inconsistent.

**Questions:**

1. The proposed IVE aims to reduce memory access. Given that it handles the general form of matrix multiplication, could it also be applied to MoE or other methods that involve extensive memory access?

---

> ### Author Response · Authors · 2024-11-19
>
> We greatly appreciate the time and effort you have taken to review our manuscript. In response to your insightful comments, we address each point individually.
>
> **For the weakness:**
> 1. **Improved Clarity and Background Information:** We acknowledge the need for a clearer presentation of complex concepts. To this end, we have revised the introduction and related work sections, redrawn figures, and included additional annotations to enhance understanding, particularly regarding the implicit value expansion technique and its relation to virtual and physical memory. Additionally, Figure 5 has been modified for better comprehension.
>
> 2. **Support for Claims in the Methods Section:**
>    - **Comparison with PKM:** In our ablation studies, UltraMem demonstrates a 0.1 improvement in loss over PKM, a gain typically associated with doubling the dense computational power. Evidence from PEER (https://arxiv.org/abs/2407.04153) showing MoE's superiority over PKM further supports our assertion that UltraMem significantly outperforms PKM even without direct comparison. The reasons for not using PKM as a baseline in larger models (151M, 680M, 1.6B) include the high training costs and inherent structural issues of PKM. PKM itself is not well-suited for scaling up training, as a very large PKM is challenging to parallelize and results in significantly longer training times.
>
>    - **Performance Over MoE:** Figures 1(b) and 1(c) in the main text illustrate UltraMem's inference speed and memory usage compared to MoE across varying batch sizes, clearly showcasing UltraMem's advantages. Tables 1 and 2 list the models' FLOPs and parameters, with UltraMem showing superior performance in benchmarks under the same activation and total parameters compared to MoE. Furthermore, we have added a new figure (Figure 6.c) comparing the inference speed of UltraMem and MoE as sparse parameters increase. This data shows MoE's inference speed growing exponentially, while UltraMem's remains almost unchanged, highlighting a significant advantage.
>
> **For the minor error:**
> 1. We thank for the detailed reading of our paper. It is indeed a typo in Eq6 where the matrix dimensions are not properly matched. We have revised the Eq6 in our latest submitted version.
>
> **For the questions:**
> 1. It's a good idea to apply IVE in MoE. Generally, memory networks and MoEs are both sparse approximations of a large inner FFN. It is possible to expand an expert to a series of virtual experts, who share the same FFN parameter while own different projectors to make them function differently. Also, the pre-pooling technique in Eq15, which saves quantities of computation, is still applicable for IVE in MoE.
>
> We hope these revisions and expansions adequately address your concerns and strengthen the paper's claims.  If there are any further questions or suggestions, please feel free to let us know. We are looking forward to providing additional clarification and discussion.

---

> > ### Comment · Reviewer_bPzv · 2024-11-24
> >
> > Thanks for your revision and explanation. I will consider whether to raise my rating after reading more discussions with other reviewers. Currently, I will keep the positive rating.

---

### Official Review · Reviewer_ch2J · 2024-10-29

**Soundness:** 3
**Presentation:** 2
**Contribution:** 3
**Rating:** 6
**Confidence:** 4

**Summary:**

In this paper, the authors propose UltraMem, a memory-efficient method designed to replace MLP layers or Mixture of Experts (MoE) in Transformer architectures, particularly for large language models. UltraMem builds on a refined Product Key Memory (PKM), utilizing a 2D grid structure with row-column key decomposition to reduce retrieval costs. Additionally, it employs Tucker Decomposition to further minimize memory usage and computational overhead. Experiments across various language tasks demonstrate UltraMem’s promising effectiveness and efficiency.

**Strengths:**

This paper addresses an interesting and impactful problem with practical applications in large language model (LLM) research.

- The authors propose an alternative, memory-efficient approach to achieving the performance of Mixture of Experts (MoE) models.
- They introduce a sparse memory access mechanism and a 2D Product Key Memory structure, which restricts memory access to only the most relevant slots, enhancing efficiency.
- **Scalability**: The use of Tucker Decomposition improves the scalability of the method, allowing it to handle larger models effectively.
- **Experiments**: The experiments demonstrate promising results, achieving comparable performance to MoE models of similar scale while maintaining greater memory efficiency.

**Weaknesses:**

Although the idea presented in this paper is novel, the experimental performance assessment shows some limitations:

1. **Lack of a Proper Baseline**: The authors do not use a well-established MoE baseline to evaluate the performance of their method. Since UltraMem is proposed as a memory-efficient alternative to MoE, it would be beneficial to compare it against real-world MoE implementations from existing open-source models, providing a more comprehensive analysis.

2. **Limited Experiments with Popular Models**: The experiments primarily use custom models, limiting the generalizability of the results. It would be valuable to assess UltraMem’s effectiveness on popular open-source models like Llama, Mistral, or others.

3. **Limited Analysis of Sparsity Levels**: The paper could benefit from a deeper investigation into the effects of different sparsity levels on memory efficiency and performance, as this is a key component of UltraMem’s approach.

**Questions:**

here are some minor writing issues, such as in Line 094, where it states 'introduce the origin large memory.' These should be easy to address.

My main questions and observations are:

1. **Comparison with Fine-Grained Mixture of Experts**: How does this method compare to fine-grained Mixture of Experts as in [1], where the hidden dimension is split between experts?
2. **Training Speed**: How does UltraMem’s training speed compare to that of MoE?
3. **Placement of UltraMem Blocks**: On what basis were the positions of the UltraMem blocks within the architecture chosen?
4. **Figure Quality**: Improving the quality of the figures and plots would enhance clarity.

**Suggestions**:

1. It would be beneficial to conduct experiments with popular open-source models like Llama-3.2-1B, Mistral, or others.
2. Since Section 3, "Why UltraMem Instead of MoE," discusses UltraMem’s advantages over MoE, it would be valuable to include experiments demonstrating that UltraMem consistently outperforms MoE in real-world scenarios, using a comprehensive baseline model for comparison.
3. Additionally, an ablation study comparing UltraMem's memory efficiency and performance against MoE based on [1] would further strengthen the analysis if applicable.

[1] *Scaling Laws for Fine-Grained Mixture of Experts*.

---

> ### Author Response · Authors · 2024-11-19
> **Authors' Response (1/2)**
>
> Thank you for your valuable and insightful feedback on our manuscript. In response to your insightful comments, we address each point individually.
>
> **For the weakness:**
> 1. Establishing a Proper Baseline with MoE: We acknowledge the importance of establishing a relevant MoE baseline for a fair comparison. It is essential to note the challenges in using existing open-source MoE models, as they often do not provide access to the training data used, or their architectural parameters differ significantly from our 1.6B UltraMem model. Therefore, we implemented a comparable MoE model with identical parameters (1.6B MoE 2in34) using the same training data to ensure a fair and meaningful comparison.
>
> 2. Experiments with Popular Models: In our paper, the 6.5B dense model has been aligned with the open-source RedPajama 7B dense model, and our evaluation metrics match those reported in their blog (https://www.together.ai/blog/redpajama-7b). This RedPajama 7B dense model replicates the results of the LLaMA 7B model. Thus, we are effectively comparing our models against established open-source models. The reason we replicated the RedPajama 7B dense model was to ensure the reliability of our training framework, as detailed in the table below.
>   | model                                 | MMLU-EM | BoolQ-EM | NaturalQuestions (closed book) - F1 | NaturalQuestions (open book) - F1 | QuAC - F1 | HellaSwag - EM | TruthfulQA - EM | avg   |
>   |---------------------------------------|---------|----------|-------------------------------------|----------------------------------|-----------|----------------|-----------------|-------|
>   | Redpajama-incite- 7b                  | 0.323   | 0.694    | 0.258                               | 0.6                              | 0.323     | 0.702          | 0.205           | 0.444 |
>   | Reproduced 7b (6.5b in paper)         | 0.246   | 0.707    | 0.272                               | 0.612                            | 0.346     | 0.691          | 0.298           | 0.453 |
>
> 3. Analysis of Sparsity Levels: We agree with your suggestion and have thus explored different sparsity levels more thoroughly. In response, we have now included additional experiments in the manuscript that compare UltraMem and MoE across various sparsity settings (Figure 6.c), enriching our analysis of how sparsity influences both memory efficiency and performance.
>   The experiments show that, for UltraMem, the inference time remains largely stable even as sparse parameters exponentially grow, provided that the activation parameters (top-m) are constant. In contrast, MoE's inference time escalates significantly under analogous conditions.

---

> ### Author Response · Authors · 2024-11-19
> **Authors' Response (2/2)**
>
> **For the questions:**
> 1. The MoE model in our paper is configured with Granularity (G=2) and Expansion (E=12), which represents a relatively strong baseline. Additionally, the Mixtral MoE employed a similar configuration.
>
> 2. UltraMem-1.6B-x12 consumed 103B tokens per day, while MoE-1.6B-x12 consumed 113B tokens per day, both on 128 A100 GPUs. Please note that while the MoE training has been extensively optimized internally, UltraMem still has room for further optimization.
>
> 3. In the original PKM paper, experimental observations suggest placing PKM in the middle of the model for optimal performance. Our early experiments confirmed that PKM achieves the best results when placed in the middle, while positioning it in the shallower layers tends to yield poorer outcomes. Consequently, for inserting UltraMem, we generally avoid the initial few layers and distribute it evenly across the subsequent layers. Regarding how many layers to skip, we primarily consider engineering aspects for training and inference acceleration. Based on our experience, we recommend skipping five layers for each UltraMem layer to approximately ensure adequate overlap for value retrieval.
>
> 4. We have redrawn all the figures to ensure the text is clear and have added annotations and references to facilitate reader understanding.
>
> **For the suggestions:**
> 1. As previously mentioned, we have aligned our studies with the open-source 6.5B dense model, thereby conducting comparisons with established open-source models. This alignment ensures that our evaluations reflect competitive benchmarks within the field.
> 2. Validating performance in real-world scenarios is indeed crucial, and we fully agree with your perspective. We have plans to train larger models and validate them in real-world scenarios after performing SFT and RLHF. In our pre-training evaluations, our benchmarks encompass a range of scenarios, including knowledge, reasoning, and reading comprehension. In these tests, UltraMem consistently matches or exceeds the performance of MoE, demonstrating its robust capabilities. This success gives us greater confidence to expand to larger models in the future.
> 3. The MoE model used in our studies already corresponds to the configuration from [1] with (G=2) and (E=12) settings. This ensures that our analysis and comparison of memory efficiency and performance between UltraMem and MoE are grounded on a well-established benchmark.
>
> We sincerely appreciate your guidance, which has helped us enhance the clarity and depth of our paper.  If there are any further questions or suggestions, please feel free to let us know. We are looking forward to providing additional clarification and discussion.

---

> > ### Comment · Reviewer_ch2J · 2024-11-24
> >
> > Thank you for conducting thorough experiments in response to my concerns. While your revisions have addressed the points raised, I am maintaining my current score pending further discussion with other reviewers. I am carefully considering whether to increase or maintain the score.

---

### Official Review · Reviewer_jFrZ · 2024-11-03

**Soundness:** 3
**Presentation:** 3
**Contribution:** 3
**Rating:** 6
**Confidence:** 4

**Summary:**

This paper proposes a novel deep learning architecture called UltraMem, designed to reduce memory access costs during the inference process of large language models, thereby improving inference efficiency. The core innovation of UltraMem lies in its introduction of an ultra-sparse memory layer, which allows the model to activate only a small number of necessary memory units when processing tasks. This approach reduces the number of memory accesses, effectively lowering inference latency. Specifically, , UltraMem surpasses MoE with the same parameters and computation as model capacity increases.

**Strengths:**

1. This paper introduces the UltraMem architecture, which demonstrates innovation by significantly reducing inference latency while maintaining computational efficiency.
2. The paper presents extensive experiments comparing the performance of UltraMem with traditional models (such as MoE and dense models), verifying UltraMem’s advantages in inference speed, memory access costs, and scalability.
3. The experiments show that UltraMem’s memory access volume grows much more slowly with batch size compared to MoE, enhancing its practical utility.

**Weaknesses:**

1. The paper lacks references and descriptions of the architecture diagrams within the main text. For example, each step in Figure 4 is not referenced in the text. Additionally, certain terms in the architecture diagrams, such as “fetch values,” are not explained in the text, making the paper difficult to follow.
2. The paper claims that the proposed UltraMem method has stronger scalability. However, UltraMem was only tested on models with 151M, 680M, and 1.6B parameters, without experiments on larger models, such as the 6.5B model or beyond. This parameter range is insufficient to fully verify UltraMem’s scalability in large-scale models. It is recommended to extend the experimental scale to explore UltraMem’s performance and scalability at extremely large parameter sizes (e.g., 10B and above).
3. Several modules proposed in the paper, such as Implicit Value Expansion (IVE) and Multi-Core Scoring (MCS), collectively enhance UltraMem’s performance. However, the independent effects of each module are not adequately evaluated. For instance, Table 2 provides ablation experiments but lacks detailed analysis of the independent effects of key modules like IVE and MCS across different tasks or sparsity settings. It is recommended to expand on Table 2 by adding experiments that assess the independent impact of each optimization module on model accuracy.
4. Figure 4 illustrates the process of Tucker decomposition but lacks analysis of how different decomposition parameters, such as rank r, affect model accuracy. To clarify the impact of different Tucker decomposition configurations on model performance, it is recommended to include more comprehensive ablation experiments to quantify Tucker decomposition’s specific role in UltraMem. Additionally, experiments should be added to analyze the effects of different values of E in the IVE method on experimental results.

**Questions:**

see weakness.

---

> ### Author Response · Authors · 2024-11-19
>
> Thank you for your valuable and insightful feedback. Below, we outline the actions we’ve taken in response to your comments, addressing each point individually.
>
> **For the weakness:**
> 1. We sincerely apologize for any inconvenience caused by our writing and figures. We have updated the manuscript to ensure that each step in Figure 4 is clearly referenced within the text. Additionally, we have enhanced the figure captions with more detailed annotations to help readers better understand the methodologies depicted.
>
> 2. We acknowledge the importance of scaling up to 10B models and certainly plan to pursue this as part of our future work. Currently, only a few studies have successfully validated models on such a large scale. In our experiments, training a 1.6B x12 UltraMem and an equivalent MoE requires 128 A100 GPUs for 10 days to process 1T tokens, whereas replicating a 6.5B dense model demands 512 A100 GPUs over approximately 14 days. While we recognize the necessity for further research in this direction, the associated computational costs remain a substantial consideration.
>
> 3. Independently ablating each modification is crucial to accurately assess the effectiveness of each component of the model, thanks very much for pointing this out. However, it is possible that applying modifications 'a' and 'b' separately on the baseline might each yield a loss improvement of -0.02, but when applied simultaneously, the combined benefit may only result in total loss improvement of -0.03 rather than -0.04. This is because individual modifications can interact and interfere with each other's effects. Therefore, we believe that sequential integration of modifications offers a more accurate understanding of their effectiveness.
>
> 4. Following your recommendation, we have included additional experiments to explore different Tucker decomposition settings (rank r = 2, 3, 4), IVE settings (E = 4, 9, 16), and MCS settings (h=2, 4, 8), which are detailed in Table 3 to better elucidate their effects on model performance.
>   The experiments show that, for IVE, as E increases, there is a consistent improvement in model performance alongside a notable increase in computational cost. However, as E rises, marginal gains decrease, leading us to recommend E=4. For TDQKR and MCS, increasing r and h does not significantly change the computational load, but the effectiveness no longer shows marked improvement, hence we suggest using r=2 and h=2.
>
> We appreciate your guidance that has significantly enhanced the depth and clarity of our paper.  If there are any further questions or suggestions, please feel free to let us know. We are looking forward to providing additional clarification and discussion.

---

> > ### Comment · Reviewer_jFrZ · 2024-11-27
> >
> > Thank you for your thoughtful response.
> > 1. The author has revised the original figures and tables, significantly enhancing the readability of the article.
> > 2. Although the computational time cost is a consideration, I believe that exploring larger-scale models is crucial for validating the proposed method.
> > 3. In my view, incorporating independent experiments to validate each component remains a more effective approach for testing the efficacy of them. This does not conflict with experiments that sequentially add modifications. Without independent validation, a new issue arises: whether the order in which components are added affects the experimental results of the method. I strongly encourage the authors to take this point into consideration.
> > 4. The authors have demonstrated the rationality of the chosen parameters by adding ablation studies for the settings in IVE, TDQKR, and MCS.
> > Overall, while the authors' response has addressed most of my concerns, the lack of well-designed ablation experiments and the absence of validation using larger-scale models make it difficult for me to assign a higher score.

---

> > > ### Author Response · Authors · 2024-11-27
> > >
> > > Thank you for your thorough review and insightful suggestions. Here are our responses to each point.
> > >
> > > Q2: Although the computational time cost is a consideration, I believe that exploring larger-scale models is crucial for validating the proposed method.
> > >
> > > A2: We recognize this as a vital step and it's indeed part of our future direction. Before proceeding further, it is crucial for us to thoroughly **understand the intrinsic nature** of the improvements that are brought about by the novel architecture we propose. As noted, we have referred to studies like Mamba[1], Jamba[2], Hymba[3], and OLMoE[4] which indicate that our current scale is already **exceptionally rare** among most structural improvements in the field.
> > >
> > > Q3: In my view, incorporating independent experiments to validate each component remains a more effective approach for testing the efficacy of them. This does not conflict with experiments that sequentially add modifications. Without independent validation, a new issue arises: whether the order in which components are added affects the experimental results of the method. I strongly encourage the authors to take this point into consideration.
> > >
> > > A3: We appreciate your suggestion about the importance of conducting independent experiments. It is indeed a valid approach.
> > >
> > >  - Rest assured, we are incorporating these ablation experiments into our subsequent results, enhancing the robustness of our validation. Before the rebuttal period concludes, we will present the results of these ablation studies. Although we will not be able to modify the PDF at that time, **we intend to incorporate the results of these independent ablations into the final version of the paper.**
> > >
> > > - It is important to emphasize that our primary goal is to **position UltraMem as a new architecture** that **outperforms existing MoE models** in both **effectiveness** and **efficiency**. Consequently, our focus has been on the incremental ablation studies, which are inherently more challenging. Many modifications in model architecture can **conflict with each other**; hence, incremental ablation is crucial. Nevertheless, we acknowledge that independent ablation studies can significantly aid in validating the effectiveness of individual modifications and can be useful in applying these techniques to other structures.
> > >
> > > We appreciate your keen insights, and they have guided us towards refining our research approach substantially.
> > >
> > > **References**
> > >
> > > [1] Gu A, Dao T. Mamba: Linear-time sequence modeling with selective state spaces. ICML, 2024.
> > >
> > > [2] Lieber O, Lenz B, Bata H, et al. Jamba: A hybrid transformer-mamba language model. ICLR, 2025, under review.
> > >
> > > [3] Dong X, Fu Y, Diao S, et al. Hymba: A Hybrid-head Architecture for Small Language Models. ICLR, 2025, under review.
> > >
> > > [4] Muennighoff N, Soldaini L, Groeneveld D, et al. OLMoE: Open Mixture-of-Experts Language Models. ICLR, 2025, under review.

---

> > > ### Author Response · Authors · 2024-12-02
> > >
> > > Considering the deadline for reviewer responses is approaching, we have predicted the results for 500B tokens based on the current progress of the ablation studies. Most of the findings are similar to those of previous incremental ablations, and the notable improvements largely remain the same six points detailed in Section 5.4 of the paper. After the experiments conclude, we will update the results into the final version of the paper. We hope that these experiments have alleviated your concerns.
> > >
> > > | Configuration             | Training Loss↓ | Validation Loss↓ |
> > > |---------------------------|---------------|-----------------|
> > > | PKM-151M-x10               | 2.604         | 2.828           |
> > > | +rm softmax               | -0.034        | -0.006          |
> > > | + half vdim+proj          | -0.028        | -0.023          |
> > > | + share query             | +0.007        | +0.003          |
> > > | + split big mem & skip    | -0.008        | -0.007          |
> > > | +query/key LN             | +0.004        | +0.001          |
> > > | +IVE                      | -0.030         | -0.025          |
> > > | +TDQKR                    | -0.003        | -0.007          |
> > > | +TDQKR + MCS              | -0.015        | -0.009          |
> > > | +value lr decay           | -0.009        | -0.015          |
> > > | + query conv              | +0.005        | +0.002          |

---

> > > > ### Comment · Reviewer_jFrZ · 2024-12-03
> > > >
> > > > Thank you for your responses, and I have no questions.

---

> > > > > ### Author Response · Authors · 2024-12-03
> > > > >
> > > > > We are pleased that our responses have addressed your questions, and we sincerely hope you might consider re-evaluating our score.

---

### Official Review · Reviewer_DbFB · 2024-11-04

**Soundness:** 4
**Presentation:** 3
**Contribution:** 3
**Rating:** 6
**Confidence:** 4

**Summary:**

Motivated by the high memory access cost bottleneck of Mixtures of Experts (MoEs) for Large Language Model (LLM) inference, the authors propose an alternative methodology, UltraMem, based on Product Ley Memory (PKM) which improves memory access above MoEs, but suffers in performance compared to them. The authors make several observations to improve the vanilla PKM methodology, basing their method on this baseline. The authors then not three problems with the PKM architecture, notably that it does not scale to large value sizes, product key decomposition biases retrieval, and unbalanced GPU computation/communication for large models. The authors propose to decompose the large memory layer of PKM into many smaller memory layers distributed across layers, allowing execution of memory and transformer layers to overlap. The authors propose to use a Tucker decomposition instead of the product key decomposition of PKM. Finally, rather than explicitly maintaining a large memory table for values, the authors propose a virtual memory approach. The authors evaluate UltraMem across several tasks compared to dense and an MoE equivalent model, and compare the performance across these tasks, along with the inference and memory accesses for each, demonstrating significantly faster inference with similar performance to an MoE model.

**Strengths:**

* Memory access is perhaps the main bottleneck in inference for contemporary hardware, and so this is a well-motivated and impactful direction to be exploring.
* The results appear to show impressive improvements over Mixtures of Experts (MoEs) in inference time and memory access while maintaining validation loss/perplexity.
* The authors methodology is detailed and thoughtful, for example deriving the correct initialization for their method (which would appear to also apply somewhat to PKM).
* Many of the figures/graphics themselves are good in design and would be helpful to understanding the methodology if it wasn't for the other serious issues with readability and captions (see below).
* Changes to PKM (before the UltraMem specific changes) are listed in 2.2 as the bag of tricks that improve PKM's performance, and are separate from the author's proposed UltraMem structure. It would be beneficial if more papers took such an approach to be clear about the differences between improved training methodology for baseline methods and a newly proposed method. Ablation in particular is nice about being clear how these changes improve baseline PKM performance.
* Real-world inference and memory access results on GPU hardware.
* Large-scale experiments, in particular results are evaluated on various model sizes, from 151M param up to 1.6B. FLOPS for the sparse models.

**Weaknesses:**

* Overall the paper reads poorly, more as a collection of experimental details and results rather than a cohesive story. I know that's frustrating and perhaps vague feedback to get as an author but I think it's really important to point out as it makes the paper quite hard to read as it is, and reduces the impact of the author's work. To be clear, I do appreciate the experimental and methodological details themselves. However, I believe the authors could do the work much more justice by taking a step back and framing their research story better. In particular, I would recommend focusing first on section 3 as higher-level motivation of the method before diving into building on top of PKM.
* All the figures containing the majority of results, and some of the methodology figures (Fig 4) are **far** too small. They are so small that the figures are completely unreadable on paper. I measured the font sizes in the figures out of curiousity, and most of the fonts are 2-3pt!
* Given that the proposed method is largely based on PKM, PKM should be a baseline in the results. The authors mention PKM's "..effectiveness is significantly inferior to that of MoE.", without explaining on what measures this is true, or giving the reader to make that judgement in the results. I don't actually doubt the authors on this, but it must be demonstrated in the experiments.
* While as noted above it's great that the "bag of tricks" for PKM is listed, along with the ablation, it also would seem important that the performance of the improved PKM should be evaluated as a baseline over just vanilla PKM, and compared with the improvements from the author's proposed method.
* The related work is **far** too short given all the work done in this field, much of which is referred to by the authors and built upon in the proposed method. To be honest I'm never a fan of related-work at the end of the paper, but I feel like this paper in particular would have greatly benefited by the related work coming before the methodology rather than at the end of the paper, as instead of having to bring up related work throughout the paper as it is built upon, it could have just been earlier summarized and then referenced throughout, allowing the story to focus on the methodology more clearly.
* Relies too much on validation loss/perplexity for the evaluation of whether generalization performance is maintained in the main paper/figures. The authors do have six other metrics for tasks in Table 1 and figures in the appendix for these tasks (which they should explain better in the paper/background rather than quickly citing all of them in a list). Given that the authors are in a sense proposing a form of sparsity, and given the results of Jaiswal et al. for pruning (Compressing LLMs: The Truth is Rarely Pure and Never Simple, ICLR 2024) I believe it's more important to focus on the performance in task-specific measures.

**Questions:**

Suggestions:
* Focus on the high-level motivation before diving into details of methodology and experiments.
* Move the related work up front in the paper, and put in background required for your method. Include MoEs, PKM, Tucker decompositions, Megatron, any background relevant to IVE and MCS, etc.
* Fix readability of all the text in the figures.
* Figure 2 (and to some extent Figure 3) are poorly labelled/captioned, and it's largely left to the reader to figure out what each side represents and how it's related to the method. At a minimum label the subfigures or describe in caption what left/right are. Would also suggest positioning floats at top of page.
* When including a large table of metrics (i.e. in Table 1), it's helpful to the reader to explain whether lower or higher is better for each, e.g. with an arrow in the header.

---

> ### Author Response · Authors · 2024-11-19
> **Authors' Response (1/2)**
>
> We greatly appreciate the time and effort you have taken to review our manuscript. In response to your insightful comments, we address each point individually.
>
> **For the weakness:**
> 1. We have revised the introduction to bolster clarity and cohesion by integrating key points from the original Section 3. This adjustment aims to elucidate our method’s motivation at an early stage, thereby enhancing overall readability.
>
> 2. Recognizing the importance of figure legibility, we have redrawn all figures to ensure that all text is clear and easily readable. Additionally, we have improved the figure captions for better comprehension. We hope these updates effectively address your concerns.
>
> 3. We acknowledge your concern and clarify that PKM was indeed used as a baseline in our ablation studies, where UltraMem showed a significant improvement in loss by 0.1. This is a considerable enhancement, usually observed with a doubling of computations in dense models. We refrained from including PKM as a baseline in our larger model trials (151M, 680M, 1.6B) primarily due to the high training costs of PKM. PKM suffers from significant memory access issues in the middle of the model, which severely reduces training efficiency. This makes it impractical to scale up the model effectively, which is one jumping-off point of our research. Specifically, training a 1.6B x12 UltraMem required 128 A100s over 10 days for 1T tokens (similar to MoE). Further tuning of PKM requires a certain amount of training cost.
>
> 4. For similar cost reasons, we had to limit the scope of certain experiments.
>
> 5. To improve logical flow, we have repositioned the related work section to precede the methodology. This restructuring allows for a more coherent narrative where the discussion of state-of-the-art techniques and their limitations in MoE and PKM provides necessary context. Furthermore, we enriched this section with a detailed discussion on Tucker decomposition and its relevance to our work, thereby facilitating better comprehension of the techniques involved.
>
> 6. Thank you for your valuable suggestions. In the evaluation section, we have categorized the downstream metrics in the paper as follows
>    -  Knowledge: MMLU, TriviaQA, GPQA, ARC
>    -  Reasoning: BBH, BoolQ, HellaSwag, WinoGrande
>    -  Reading Comprehension: DROP
>    -  Overall Performance: AGIEval
>
>    These downstream metrics cover the categories of benchmark tasks discussed in Jaiswal's work, including both zero-shot and few-shot (in-context learning) settings. Furthermore, since we are focusing on the pre-trained models (without alignment), the instruction-following setting in Jaiswal's paper may not be the most suitable evaluation setting for our work.
>
>    Regarding your point about using loss and perplexity to measure a model's general capabilities, it is indeed an interesting question! In general, as mentioned in Jaiswal's paper, during inference, "...all compressed models are derived from the same dense counterpart with high similarity, and aforementioned differences(scale, training strategy, architecture) don’t exist, making their evaluation more challenging". However, in our case, we are focusing on pre-training (training models from scratch), and the models with different sparsity levels are non-homogeneous (e.g.,  the cosine similarities of the same modules' weights among them are less than 0.2). In such cases, the training/validation loss will be an efficient and reliable indicator of general performance, as the breadth of the training/validation corpus will far exceed that of specific task datasets. In fact, when exploring the scaling laws of pre-training, the validation/test loss is commonly used as a metric for measuring a model's general capabilities, as seen in the paper of OpenAI's scaling law (https://arxiv.org/abs/2001.08361) and the technical report of LLaMA-3 (https://arxiv.org/abs/2407.21783).

---

> ### Author Response · Authors · 2024-11-19
> **Authors' Responses (2/2)**
>
> **For the suggestions:**
> Thank you for your detailed suggestions. We have taken your feedback into careful consideration and have made the following revisions to our manuscript:
>
> 1. We have rewritten the introduction to focus on high-level motivation, clearly outlining the current issues, what our methodology aims to address, and the extent of its capabilities.
>
> 2. The section on related work has been moved to the front of the paper as the second section. We've enriched this part with detailed background information on MoEs, PKM, Tucker Decomposition to provide readers with a thorough understanding of the current research landscape. Megatron is not central to our research direction; therefore, it is omitted from the related work section.
>
> 3. All figures have been redrawn to ensure all text is clearly visible. Additional annotations have been included to aid in understanding.
>
> 4. We have improved the captions and labeling in Figures 2 and 3, providing detailed descriptions of what each side represents and how it relates to the method. Subfigures are now clearly marked, and we have adjusted the positioning of floats to appear at the top of the pages.
>
> 5. We have included arrows in the headers to indicate whether higher or lower values signify improvement, aiding in clearer interpretation of the metrics.
>
> We hope these adjustments meet your expectations and contribute to the improved readability and comprehension of our paper. If there are any further questions or suggestions, please feel free to let us know. We are looking forward to providing additional clarification and discussion.

---

> ### Comment · Reviewer_DbFB · 2024-11-23
>
> First, I would like to thank the authors for their revised paper and rebuttal explaining the changes/what they addressed, it's clear they have taken the feedback seriously and made a significant effort to effect those changes within the rebuttal period.
>
> Quickly reviewing the revised paper, and going through the weaknesses I identified in the initial version:
> - The figures are much improved, now legible, and in some cases with better captions, although some of the later figures could also do with better captions still.
> - I like the new text for the introduction, it much better motivates the method - especially in explaining the inference issues with MoEs!
> - Moved background to before method, which I think works much better for explaining the author's method
> - Expanded background section with tensor decomposition serves the authors' work much better.
> - Improved results table with arrows indicating if metric is desired to be small/large
>
> Further changes I think would help the author's work:
> - Expand the caption in Figure 4/5, they are complex figures, and now I can read it properly, I found the caption very brief and not helpful. Ideally you want your figures to be standalone (as much as that's ever possible).
>
> Authors rebuttal for PKM baseline/limitations of experiments due to compute (Weakness 3/5):
> I think this is, to some degree, a fair point: that PKM is a costly baseline to run for the larger models and that's what motivated the research, and I'm with the authors on that motivation completely. Also in general, as much as possible as a reviewer I avoid demanding prohibitively expensive experiments to keep research accessible. However, in the context of MoE research, all experiments are going to be very computationally expensive, and specifically in your case you are trying to motivate claims are you are doing better than the relatively expensive PKM so this seems to me to be a minimum requirement to have those complete baselines to me. In summary, while I do understand this is all very computationally expensive, if we let papers make claims with MoEs specifically go unverified because experiments are expensive, it would be bad for the research community in the long run. Also want to be clear I wouldn't expect all such results to be feasible in the rebuttal period, but was hoping perhaps for some commitment to have them by final paper.
>
> Summary:
> Overall I am much happier with the revised paper and I do think it addresses a lot of the low-hanging issues with the initial submission on the layout of the paper, figures, and much (but not all) of the writing, and will be re-evaluating my rating of the work in that context. However, while I sympathize with the computational issues, I believe the lack of some key baselines is a major issue still that restrains me from giving a much stronger rating.

---

> > ### Author Response · Authors · 2024-11-28
> >
> > Thank you once again for your valuable suggestions, with which we completely agree. Consequently, we have made the following amendments to our paper:
> >
> > 1. We have enriched the captions of Figures 4 and 5 to better assist readers in understanding our modification strategy directly from the images.
> > 2. We have initiated experiments for PKM models at 151M, 680M, and 1.6B. Some preliminary results have already been updated in the paper. Once the models complete training, we will update the paper's final version with the end results, which we anticipate will show PKM significantly underperforming compared to UltraMem.
> >
> > We are truly grateful for all your suggestions, which were indeed apt and have made this paper more comprehensible and solid. We hope these changes have alleviated your concerns.

---

> > ### Author Response · Authors · 2024-12-02
> >
> > Considering that the deadline for reviewer responses is imminent, we have recalibrated our final metrics predictions based on our current training progress. Currently, PKM-680M-x12 is 80% trained, while PKM-1.6B-x12 is at 40%. The table below displays our recalibrated predictions. The asterisk means the result is predicted. The performance of PKM is less effective than both MoE and UltraMem, and due to its lack of large-scale training capabilities, this set of experiments tends to train slower compared to MoE and UltraMem. We trust that these experimental results will address your concerns, and we would be grateful if you could kindly reconsider our score.
> >
> >
> >
> >
> >
> > | Model              | Param (B) | FLOPs (G) | Val. loss↓ | GPQA↑ | TriviaQA↑ | BBH cot↑ | HellaSwag↑ | WinoGrande↑ | DROP↑  | Avg↑   |
> > |--------------------|-----------|-----------|-----------|------|----------|---------|-----------|-------------|-------|-------|
> > | Dense-151M         | 0.15      | 0.30      | 2.96      | 19.98| 12.67    | 22.57    | 35.07     | 52.49       | 13.60 | 26.06 |
> > | ***PKM-151M-x12***       | 2.04      | 0.35      | 2.76      | 17.30| 24.66    | 23.14    | 42.25     | 51.38       | 13.10 | 28.64 |
> > | MoE-151M-2in32     | 2.04      | 0.35      | 2.63      | 17.30| 33.27    | 23.24    | 48.44     | 55.96       | 18.57 | **33.20** |
> > | UltraMem-151M-x12  | 2.03      | 0.35      | 2.67      | 19.42| 28.97    | 22.65    | 43.96     | 50.83       | 14.08 | 29.99 |
> > | Dense-680M         | 0.68      | 1.36      | 2.64      | 21.09| 27.16    | 24.65    | 48.83     | 54.93       | 22.97 | 33.27 |
> > | ***PKM-680M-x12****     | 8.95      | 1.50      | 2.45      | 19.31| 42.75    | 26.04    | 57.69     | 58.52       | 23.73 | 38.01 |
> > | MoE-680M-2in33     | 8.95      | 1.50      | 2.39      | 20.54| 34.19    | 26.63    | 62.71     | 59.98       | 26.54 | 38.43 |
> > | UltraMem-680M-x12  | 8.93      | 1.49      | 2.37      | 21.99| 55.17    | 26.62    | 64.15     | 60.54       | 25.14 | **42.27** |
> > | Dense-1.6B         | 1.61      | 3.21      | 2.49      | 21.76| 39.65    | 26.41    | 58.60     | 61.72       | 22.63 | 38.46 |
> > | ***PKM-1.6B-x12****      | 21.13     | 3.48      | 2.32      | 23.99| 56.54    | 28.45    | 66.63     | 65.83       | 29.04 | 45.08 |
> > | MoE-1.6B-2in34     | 21.36     | 3.52      | 2.30      | 21.32| 59.56    | 29.46    | 67.34     | 63.93       | 28.81 | 45.07 |
> > | UltraMem-1.6B-x12  | 21.41     | 3.50      | 2.24      | 24.66| 66.38    | 30.63    | 71.52     | 66.38       | 29.99 | **48.26** |
> > | Dense-6.5B         | 6.44      | 12.88     | 2.30      | 19.98| 57.28    | 31.14    | 69.73     | 65.9        | 33.12 | 46.19 |

---

### Author Response · Authors · 2024-11-19
**General Response**

We extend our deepest gratitude to all the reviewers for their time and valuable feedback. We have carefully considered your comments and have made the following revisions to the manuscript, highlighted in red for your convenience:
1. We have rewritten the introduction to provide a higher-level analysis of the causes behind UltraMem, facilitating a clearer understanding of our motivation.
2. The related work has been enriched and moved to Section 2. In this section, we have highlighted points pertinent to our study, making it easier for the readers to grasp the prior work and the key challenges in this field.
3. All the reviewers pointed out issues with the images. We have redrawn all the images to ensure text clarity, added captions, and updated the corresponding citations in the manuscript to aid reader comprehension.
4. Minor typos in the equations have been corrected, and we have provided explanations for previously undefined concepts, such as physical memory and virtual memory.
5. We have moved some improvements related to Megatron to the appendix due to the extensive amount of content in the main text and its lower relevance to the core contributions of this paper.
6. Clarifications on the specific benchmarks used have been added. Our benchmarks encompass knowledge, reasoning, reading comprehension, and comprehensive ability.
7. A new comparative experiment between UltraMem and MoE under varying sparsity parameters has been added. The results indicate that, with continuous increases in sparse parameters while keeping activation parameters constant, UltraMem's inference time does not experience significant growth, whereas MoE's inference time grows exponentially.
8. We have added three sets of ablation experiments to assess the impact of IVE, TDQKR, and MCS under different hyperparameters, demonstrating the rationale behind our choices of E=4, r=2, and h=2.
In addition to submitting the revised manuscript, we will also provide individualized responses to each reviewer to address their specific concerns and questions. We hope that these revisions and responses will clarify any ambiguities, address all the reviewers' concerns, and lead to reconsidering increasing the rating for our manuscript.


**References**

[1] Jegou H, Douze M, Schmid C. Product quantization for nearest neighbor search. IEEE transactions on pattern analysis and machine intelligence, 2010.

[2] Roller S, Sukhbaatar S, Weston J. Hash layers for large sparse models. NeurIPS, 2021.

[3] Geva M, Schuster R, Berant J, et al. Transformer feed-forward layers are key-value memories. EMNLP, 2021.

[4] Chi Z, Dong L, Huang S, et al. On the representation collapse of sparse mixture of experts. NeurIPS, 2022.

[5] Zhou Y, Lei T, Liu H, et al. Mixture-of-experts with expert choice routing. NeurIPS, 2022.

[6] Krajewski J, Ludziejewski J, Adamczewski K, et al. Scaling laws for fine-grained mixture of experts. ICLR, 2024.

[7] Bershatsky D, Cherniuk D, Daulbaev T, et al. LoTR: Low tensor rank weight adaptation. arXiv, 2024.

---

### Author Response · Authors · 2024-11-22

Many thanks for your comments and acknowledgment of our work. If you have any additional comments or suggestions, we sincerely look forward to the opportunity to engage in further discussions with you and are more than happy to revise our manuscript further.

---

### Author Response · Authors · 2024-11-28
**General Response 2**

Thank you to all the reviewers for your invaluable comments and suggestions. Based on your feedback, we have made the following revisions to our paper:

1. We have conducted experiments for PKM-151M, 680M, and 1.6B, which serve as baselines for UltraMem. Currently, PKM-151M-x12 has completed its evaluation. PKM-680M-x12 is 30% through its training, and based on available results, we have predicted its performance at the end of training, which is now included in Table 1. PKM-1.6B-x12 is still in the early stages of training, and thus reliable results are not yet available; these will be updated in the final version of the paper. Table 6 has been supplemented with the hyperparameter configurations of the PKM models, and Figure 11 has been updated to include the current evaluation curves for PKM.

2. We have conducted additional ablation studies, specifically implementing various modifications independently on the PKM structure to assess the benefits of each change. We are currently unable to compile the results, but they will be added in the coming days and reflected in the final version of the paper.

Once again, we thank the reviewers for your valuable suggestions. These iterations have made the conclusions of Ultramem more robust, and we hope the revisions will prompt reconsideration of our score.

---

### Author Response · Authors · 2024-12-04
**Final Author General Response**

Dear Reviewers and Area Chairs,

We sincerely appreciate the attention to detail and valuable suggestions offered by the reviewers throughout the review process, as well as the guidance provided by the Area Chairs. As we approach the end of the rebuttal phase, we would like to summarize the core contributions of our manuscript, "Ultra-Sparse Memory Network," and reflect on the interactions we've had concerning the feedback provided.

### **Main Contributions**

Our paper, "Ultra-Sparse Memory Network," introduces an ultra-sparse memory layer to reduce memory access costs during LLM inference, stands as a significant advancement. This not only preserves the performance levels of conventional models, such as Mixtures of Experts (MoE), but also surpasses them in terms of inference speed and **scalability**, while maintaining similar training speed compared to MoE. Our submission highlights the following key advancements:
1. **Efficient Memory Access Reduction**: We effectively demonstrate how UltraMem decreases the required memory access for model operations, leading to significant improvements in inference speeds.
2. **Maintenance of Performance While Scaling**: Similar to traditional dense models and existing sparse models like MoE, our method maintains competitive performance metrics even as the model scales.
3. **Superior Inference Speed and Scalability**: During ongoing scaling, UltraMem's performance matches or exceeds those of MoE, while maintaining a significant speed advantage in inference, ranging from 2 to 6 times faster.

These contributions were positively acknowledged as pioneering by the reviewers:
- "…an impressive approach to reducing computational load while maintaining strong model performance…" (Reviewer DbFB)
- "...real-world inference and memory access results on GPU hardware..." (Reviewer DbFB)
- "…a significant step forward in addressing LLM inference bottlenecks…" (Reviewer jFrZ)
- "...achieving comparable performance to MoE models of similar scale while maintaining greater memory efficiency..." (Reviewer ch2J)
- "...significantly outperforms existing approaches on large models, being up to six times faster than MOE at the same scale...." (Reviewer bPzv)

---

> ### Author Response · Authors · 2024-12-04
> **Final Author General Response 2**
>
> ### **Summary of Revisions**
>
> In response to the constructive feedback from the reviewers, we have made several significant adjustments and additions:
>
> **For common response:**
> 1. **Improved Logical Flow**: Following Reviewer DbFB’s insightful suggestions, we restructured our paper to enhance coherence. We moved the related work to the beginning to detail MoE and PKM limitations and expanded the discussion on Tucker decomposition. Additionally, we refined the introduction to incorporate essential insights earlier, enhancing clarity and readability.
> 2. **Enhanced Clarity and Detailing**: In response to all reviewers, we significantly improved the manuscript's clarity, especially around the technical descriptions of our architecture and the experimental setup, ensuring that our methods are understandable and replicable.
>
> **For individual response:**
> 1. **Reviewer DbFB** expressed concern about the lack of a fundamental baseline, specifically the performance of the PKM model at the scales of 151M, 680M, and 1.6B. Upon reevaluating our experiments, we agreed with the reviewer's perspective. During the rebuttal period, **we conducted these three experiments**, making the comparison with UltraMem more solid.
> 2. **Reviewer jFrZ** noted that our ablation studies were missing certain verificatory experiments. These include evaluating the performance of IVE, TDQKR, and MCS under various configurations and assessing the benefits of ablating each component separately. We fully agree with this observation. Consequently, we conducted four additional sets of ablation experiments. These experiments were designed to **validate the optimal configuration for IVE, TDQKR, and MCS**, as well as to **independently ablate each component**. The outcomes of these ablations robustly **support our model structure and hyper-parameter configurations**.
> 3. **Reviewer ch2J** was concerned about the lack of comparisons with public models and the absence of inference speed evaluations at different sparsity levels. We addressed the first concern by comparing our dense baseline (Dense-6.5B) with the open-source model Redpajama-incite-7b in our paper, showing similar performance, which **confirms our model's validity against public models**. Furthermore, we conducted extra experiments comparing the inference speeds of UltraMem and MoE when increasing sparsity parameters with fixed activation parameters. These ablations consistently demonstrated UltraMem's superiority, showing **minimal impact on its speed with increased sparsity**, unlike **MoE**, which **significantly slows down**.
> 4. **Reviewer bPzv** felt that comparing UltraMem and PKM solely based on validation loss was not sufficiently convincing and that comparisons with MoE lacked detailed metrics such as memory access, FLOPs, and parameter counts. Upon re-evaluating our experiments, we agreed with the need for more extensive PKM comparisons. Consequently, we trained PKM models at scales of 151M, 680M, and 1.6B. **Evaluating these against UltraMem, the results further validated UltraMem's effectiveness**. Regarding the missed comparisons, we had actually detailed these metrics in the main text of the original manuscript, which might have been overlooked due to clarity issues. **The revised version of the paper now clearly highlights these comparisons** to ensure better understanding.
>
> We believe that the majority of the reviewers' concerns have been addressed through additional ablation experiments, improved clarity of expression, and detailed explanations. These enhancements, driven by reviewers' feedback, have significantly fortified the technical solidity and clarity of our paper.
>
> We are grateful once more for the engaging and constructive review process, which has undeniably enriched our work.
>
> Best regards,
>
> Authors of UltraMem

---

### Meta-Review · Area_Chair_8Wpw · 2024-12-21

**Metareview:**

The Ultra-Sparse Memory Network (UltraMem) paper presents a significant advancement in the efficiency and scalability of large language models (LLMs) by introducing an ultra-sparse memory layer. This approach addresses the memory access cost bottleneck in LLMs, particularly during inference, where it outperforms the Mixture of Experts (MoE) models in terms of speed, ranging from 2 to 6 times faster. The reviewers have noted several key strengths and weaknesses in the submission.

Strengths:
* Efficiency and Scalability: UltraMem demonstrates impressive improvements in inference speed and memory access efficiency while maintaining model performance. This is a critical advancement for practical applications of LLMs.
* Methodological Approach: The authors' methodology is detailed and thoughtful, with significant improvements to the baseline Product Key Memory (PKM) model. The paper's approach to clearly delineating changes to the baseline and the newly proposed method is commendable.
* Experimental Rigor: The submission includes extensive experiments, including comparisons with dense models and MoE, validating UltraMem's advantages across various tasks and model sizes.

Weaknesses:
* Presentation and Readability: The initial version of the paper was criticized for poor readability and a lack of cohesion. However, the authors have made substantial revisions to address these issues.
* Baseline Comparisons: The lack of direct comparisons with PKM in larger model sizes and the absence of comparisons with popular open-source models were noted as limitations. However, the authors have since initiated experiments to address these concerns.
* Ablation Studies: Some reviewers called for more comprehensive ablation studies to independently validate the effectiveness of each proposed modification. The authors have responded positively, committing to include these studies in the final version of the paper.

Based on the reviewers' feedback and the authors' responses, I recommend accepting the "Ultra-Sparse Memory Network" paper for presentation at the ICLR 2025 conference. The paper introduces a novel and impactful approach to improving LLM efficiency, backed by rigorous experimentation and a clear commitment to addressing reviewer feedback. The revisions have significantly improved the paper's presentation and addressed the primary concerns raised by the reviewers, positioning UltraMem as a valuable contribution to the field of large language model research.

**Additional Comments On Reviewer Discussion:**

Points Raised by Reviewers:

* DbFB: Poor readability, lack of PKM baseline, insufficient related work, and over-reliance on validation loss.
* jFrZ: Lack of references for architecture diagrams, limited scalability tests, and inadequate ablation studies for individual modules.
* ch2J: Lack of proper MoE baseline, limited experiments with popular models, and insufficient analysis of sparsity levels.
* bPzv: Need for clearer explanation of concepts, insufficient direct comparison with PKM and MoE, and a typo in Equation 6.

Authors' Responses:

* DbFB: Revised introduction and related work, redrawn figures with improved captions, initiated PKM experiments at larger scales.
* jFrZ: Updated figures and text, committed to larger-scale model experiments and independent ablation studies.
* ch2J: Implemented MoE baseline, aligned with open-source models, added experiments on different sparsity levels.
* bPzv: Clarified concepts, added direct comparisons with PKM and MoE, corrected Equation 6 typo, and discussed IVE applicability.

---

### Decision · Program_Chairs · 2025-01-22

Accept (Poster)